# Multi-sectoral and systemic drought risk in forested cold climates: Stakeholder-informed vulnerability factors from Sweden

Elin Stenfors[1], Malgorzata Blicharska[2], Thomas Grabs[1], Claudia Teutschbein[1]

1   Uppsala University, Department of Earth Sciences, Program for Air, Water and Landscape Sciences, Villavägen 16, 75236 Uppsala, Sweden

2   Uppsala University, Department of Earth Sciences, Program for Natural Resources and Sustainable Development, Villavägen 16, 75236 Uppsala, Sweden

*Correspondence to*: Elin Stenfors (elin.stenfors@geo.uu.se)

**Abstract.** There is a global call for proactive drought risk management, stressing the need to further our
understanding of the systemic nature of drought risk. Proactive drought risk management requires not only an
understanding of the drought hazard itself, but also of the underlying vulnerabilities in socio-hydrological systems.
As a result, drought vulnerability assessments are increasingly conducted across the globe. However, drought
vulnerability is complex and shaped by the social, ecological and hydroclimatic context. Thus, understanding how
vulnerability is manifested depending on regional, sectoral or societal differences is crucial. Therefore, we here
present an assessment of the practical relevance and relative impact of various drought vulnerability factors for
water-dependent sectors and societies in forested cold climates. The analysis was based on the results of an online
survey conducted in Sweden, targeting stakeholders from seven water dependent sectors, working in authorities,
private and public enterprises, NGOs and trade associations. Respondents were asked to rate a comprehensive list
of vulnerability factors, connected to sectoral and societal vulnerability as well as governance, based on their
perceived impact on drought risk in their sector as well as for society as a whole. Results showed that the relevance
and impact of individual vulnerability factors differed across sectors, with the forestry sector especially standing
out compared to other sectors. Furthermore, the results indicate regional differences in societal vulnerability
factors. The substantial list of vulnerability factors found to be relevant by the respondents demonstrates the
complex nature of drought risk, as well as the importance of adopting cautiousness when selecting generic
vulnerability factors for applied vulnerability assessments. Furthermore, the results provide a comprehensive
guide to both sectoral and societal drought vulnerability in socio-hydrological systems located in forested cold
climates.

## 1  Introduction

In recent years, Europe has experienced large scale drought events, the most recent in 2022 when the continent faced unprecedented drought conditions (Faranda et al., 2023), and the 2018-2019 drought that affected large parts of Europe including the high-latitude regions of Scandinavia (Bakke et al., 2020; Teutschbein et al., 2022). Droughts are slow on-set and recurrent phenomenon (Wilhite, 1996) capable of affecting various aspects of socio-hydrological systems. Brought on by climatic variability that creates regional water deficits compared to normal conditions, droughts can manifest in all climatic zones (Wilhite, 1996). They are generally classified into four drought types based on where in the hydrological system they arise; meteorological (precipitation deficit sometimes combined with evapotranspiration), soil moisture (soil moisture deficit), hydrological (negative anomalies in surface or groundwater), and socioeconomic drought (impacts on water as an economic good) (Mishra and Singh, 2010; Van Loon, 2015; Wilhite and Glantz, 1985). Many sectors in society are dependent on water. The interconnectedness and interdependencies between sectors as well as society can give rise to wide ranging drought impacts with cascading and ripple effects (Hagenlocher et al., 2023; UNDRR, 2021). Consequently, there is a growing call for understanding the systemic nature of risk (UNDRR, 2019), incorporating multi-sectoral and whole-of-society systems perspectives (Hagenlocher et al., 2023; Stefanski et al., 2025). Such endeavors require an understanding, not only of the drought hazard itself, but of the underlying vulnerabilities of a system, that can exacerbate the impacts of a drought. Analyzing and understanding drought vulnerability can improve drought risk management and the resilience of socio-hydrological systems. There is currently no interdisciplinary consensus on how to define and conceptualize drought vulnerability (Ciurean et al., 2013; Fuchs and Thaler, 2018; Jamshed et al., 2023; Rufat and Metzger, 2024; Turesson et al., 2024). It can broadly be explained as an entity's predisposition to drought-related harm (Füssel, 2007; Turner et al., 2003), influenced by the drought type, its duration (Adger, 2006) and location (Turner et al., 2003). Consequently, indicators used to measure vulnerability will depend on factors like management practices, cultural context and historical hydrological conditions (Taylor et al., 2009), with the importance of each factor differing based on the specific hydroclimatic and social conditions of the area (McEwen et al., 2021). Yet, indicator-based assessments often use the same set of vulnerability indicators, independent of the context of the assessment (Hagenlocher et al., 2019).

Due to the contextual nature of drought vulnerability, Stenfors et al. (2024) presented a conceptual model for drought vulnerability in forested cold climate regions, based on a substantial list of vulnerability factors that influence drought risk in the climate and eco-region. The model divides vulnerability factors into three distinct categories: (1) direct water consumers, i.e., factors linked to sectors or groups using water directly (e.g., for drinking water, watering crops, etc.), (2) indirect water consumers, i.e., factors related to societal groups that use water indirectly through consumption of goods that need water for production (e.g., food, energy, etc.), and (3) governance processes and plans, such as policies and plans concerning drought, financial ability to adapt or respond to drought, where factors connected to the category can affect both sectors and society as a whole. Their literature review identified more than 80 factors relating to drought vulnerability in the climate region. However, the practical relevance of the identified factors for practitioners, as well as the relative impact of the vulnerability factors within and across different societal sectors is still unknown. For example, one of the most commonly used vulnerability factors in the reviewed literature is irrigation. Meanwhile, only approximately 3% of Swedish agriculture is irrigated (The Swedish Board of Agriculture, 2024) which raises the question on the relevance of this factor in a local setting. Hence, further work is required to assess these factors in order to better understand

their systemic relevance and relative impact on multi sectoral and societal drought risk in forested cold climates. Such information is valuable when designing vulnerability assessments in the climate region, as common approaches often involve combining several vulnerability factors into a vulnerability index, used to assess the vulnerability of a study region(Blauhut et al., 2016). Index-based vulnerability assessments often incorporate vulnerability factors either by assigning them equal weights (Fekete, 2012; Tate, 2012), indicating that each factor has the same influence on drought risk (Tate, 2013), or by assigning weights to capture the relative importance of each factor on the vulnerability profile of the study area (Engström et al., 2020; Meza et al., 2020; Ortega-Gaucin et al., 2018). Whilst index-based approaches are a simplification of the complex and dynamic nature of vulnerability, it has been shown that assigning weights to the incorporated vulnerability factors, can improve the precision and robustness of the assessments (Martín et al., 2017; Moshir Panahi et al., 2023) and provides a deeper understanding of societal heterogeneities (Moreira et al., 2023) and local specificities(Martín et al., 2017). A commonly used method for assigning weights is to incorporate expert knowledge through stakeholder engagement. For example, Martín et al. (2017) involved an expert panel consisting of local practitioners for adjusting indicator weights to their community vulnerability index for natural hazards, and Hung and Chen (2013) used an analytic network process involving focus group meetings for assessing vulnerability to climatic hazards in a river basin in Taiwan. Similarly, Meza et al. (2019) used a survey study to assess the relative importance of drought vulnerability indicators for agriculture and water supply on a global scale. The importance of vulnerability factors vary depending on study region and incorporating participatory weighting techniques offer valuable insights into the distinct characteristics of an area or region (Moreira et al., 2023). Hence, expert knowledge and stakeholder perceptions can help identify drought vulnerability factors in forested cold climates specifically adapted to stakeholder needs in different socio-economic sectors. Furthermore, index-based approaches incorporating multisectoral or holistic perspectives usually focus on a combination of sectors such as agriculture, water supply, energy, and industry (Engström et al., 2020; Kim et al., 2015; Radeva and Nikolova, 2020; Shiravand and Bayat, 2023; Wang and Sun, 2023), often failing to incorporate the forestry sector which is an important sector in forested climate regions and forestry-dependent economies. Meanwhile, Melo et al. (2020) and Tidwell (2016) argued for the importance of forests for water, energy, and food securities and the need to update nexus perspectives to include a forest dimension when analyzing multi-sectoral vulnerability to natural and socio-economic hazards (Melo et al., 2020).

This paper aims to fill this gap and identify the most influential drought vulnerability factors for sectoral and cross sectoral vulnerability in forested cold climates, thereby informing applied vulnerability assessments, integrated drought risk management efforts, and whole-of-society adaptation efforts. Using the conceptual framework proposed by Stenfors et al. (2024), the research objectives for this paper are to (1) assess the sectoral and cross-sectoral relevance of previously identified drought vulnerability factors for water-dependent sectors as well as society in forested cold climates, (2) determine their relative rankings through the use of impact scores, (3) identify the highest-rated vulnerability factors for sectoral and cross sectoral and societal vulnerability, (4) explore variations in ratings among the respondents, hypothesizing that impact ratings would vary based on stakeholder's (i) sectorial focus or type of organization, (ii) geographical location, (iii) level of drought experience, and lastly to (5) improve the current understanding of drought vulnerability by presenting newly discovered vulnerability factors reported by the respondents.

## 2 Methods

### 2.1 Study area

This study focuses on Sweden in northern Europe. With a population of 10.5 million people over a land area of approximately 408,000 km$^2$, Sweden has an average population density of approximately 25,8 inhabitants per square kilometer where the northern inland areas are much less populated compared to the southern and coastal areas of the country. Forestry and agriculture make up 2,5% and 1,3% of Sweden's GDP respectively. Energy production is made up of nuclear (30%), hydropower (35-45%), and wind power (18-20%).

Sweden is divided into three climate zones according to the Köppen-Geiger classification (Beck et al., 2018). The climate ranges from tundra (ET) in the Scandinavian Mountains in north-western Sweden with monthly mean temperatures below 10 °C, subarctic boreal (Dfc) climate with cool summers, very cold winter, and seasonal snow cover and soil frost during winters in central and northern Sweden, and a warm-summer hemi boreal (Dfb) climate zone in southern Sweden. Most areas currently classified as Dfb and Dfc climate zones are projected to shift into Cfb and Dfb climates respectively by 2070–2100 (Beck et al., 2018). A majority of Sweden's land area is covered by forests (69%), followed by wetlands (9%), shrubs and grassland (8%), agriculture (8%), human settlements (3%) and open land (3%) (SLU, 2015). It has historically been seen as a country with abundant water resources, with an average annual precipitation of 784 mm during the period of 1961-2020 combined with low evapotranspiration. The mean annual temperature during the period was 2.6 °C, with an increasing temperature corresponding to 0,037 °C per year or a total warming of 2.2 °C during the observation period (Teutschbein et al., 2023b).

Sweden has three levels of government: national, regional and local. On a regional level, Sweden is divided into 21 counties whose political tasks are divided between regional councils and county administrative boards. Regional councils comprise county-elected decision-makers, while the county administrative boards are government bodies within the counties. At the local level, Sweden has 290 municipalities, each with an elected municipal council that handles municipal decision making. From a water management perspective, Sweden is divided into five water districts, based on the bounds of major sea basins and catchment areas. As a result, the regional and local authorities can be part of more than one water district. Each water district is appointed one of the county administrative boards to act as the water district authority. The water district authority manages the aquatic environment in the water district by, for example, preparing management and action plans, coordinating water management work on county administrative boards and municipalities, and collaborating with authorities and other interested bodies on national to local level. Local authorities are responsible for providing water supply, either directly or through municipally owned water enterprises. According to Statistics Sweden (2022), a large majority of Swedish households (87%) are connected to public drinking water networks, where approximately 51% comes from surface water. However, there are regional differences in household connectivity to public drinking water, ranging from 69% in Gotland to 94% of Stockholm county (Statistics Sweden, 2022).

The agricultural sector accounts for four percent of the total freshwater water use in Sweden, with large regional variations (Vattenuttag, 2024). Freshwater use in the agricultural sector mainly comprises crop irrigation and drinking water for animals. The majority of irrigation water is used in the southern-most county Skåne in Sweden (56% of total water use for irrigation) that accounts for 41% of the irrigable area (i.e. the maximum area that can be irrigated using available equipment and water) in Sweden (Statistics Sweden, 2022). However, there are large uncertainties regarding the amount of water used for irrigation as well as its water source. Surveys conducted in

the 1970s and 1980s, showed that 85% of irrigation water use came from surface waters and the rest was mainly
from private groundwater aquifers. During 2020, industrial water use accounted for 2 097 million cubic meter of
water use, out of which 47% was used as cooling water in electricity production. There are three water intensive
industries that account for approximately 80% of the total industrial water use: paper & pulp industry, chemical
production, and steel and metal works (Statistics Sweden, 2022).
The 2018 drought had several impacts on Sweden. The combination of high temperatures and low precipitation
gave rise to hydrological and agricultural droughts in several parts of the country  (Sjökvist et al., 2019; Stensen
et al., 2019). It impacted sectors including energy, agriculture, water, and forestry, along with the environment,
and resulted in various societal effects (Sjökvist et al., 2019). Crop yields for a variety of crops were halved
compared to the five-year average (Lantmännen, 2018), resulting in an estimated loss of more than 900 million
USD[1] in irretrievable harvests (LRF, 2019). Furthermore, harvest losses and dry conditions created a pasture and
fodder shortage for farm animals and emergency slaughter increased drastically with waiting times being up to
six months long (Sjökvist et al., 2019). Consequently, availability of grains, dairy product and meat was reduced.
The drought impacts on the hydrological system also affected several sectors. Inflow to hydropower reservoirs
was exceptionally low, which ultimately created a 50-70% rise in electricity prices during the summer of 2018
(Sjökvist et al., 2019). Several Swedish municipalities saw water shortages, where 85 municipalities introduced
restrictions on irrigation and 100 municipalities urged its inhabitants to lower their water consumption
(Krisinformation, 2018). The reduced water flows also had an impact on ecosystems. For example, water courses
housing important nursery habitats for salmon and sea trout dried up (S. V. T. Nyheter, 2018). The dry conditions
of 2018 also resulted in forest fires over large areas in Sweden. Forest resources worth 84 million USD[1] were lost
due to the fires (Sjökvist et al., 2019). Several Swedish counties were affected and in total 25 000 forest hectares
were lost, with more than 500 individual forest fires identified during the period (MSB, 2018).
**2.2    The original drought vulnerability framework**
Based on the conceptual framework described by Stenfors et al. (2024), vulnerability factors can be divided into
three categories connected to the attributes of (1) direct water consumers (here-after: sectoral factors), i.e., groups
or sectors that use water directly (e.g., irrigation or drinking water), (2) indirect water consumers (societal factors),
which consist of groups or sectors that use water indirectly by consuming goods that require water for their
production (e.g., food or energy), (3) governance processes and plans (governance), that is governing processes,
policies, tools, and plans that affect a sector or society's ability to cope and adapt to drought. As such, governance
factors can affect the vulnerability of both individual sectors and society as a whole.
The conceptual model adheres to the IPCC AR6's (IPCC, 2022) definition of vulnerability, i.e., *"vulnerability*
*encompasses a variety of concepts and elements including sensitivity or susceptibility to harm and lack of capacity*
*to cope and adapt". S*usceptibility is an elements' predisposition to harm by an external or internal stressor, coping
capacity is its ability to react and respond to a stressor and adaptive capacity is its ability to learn from past
stressors and anticipate future stressors. The conceptual model was developed on the basis of a literature review,
identifying vulnerability factors studied or applied in countries with forested ecoregions and cold or continental
climates. After analysis, the 83 identified vulnerability factors were divided into those relating to *sectors* as direct
water consumers (51 factors), *society* as indirect water consumers (9 factors), and *governance* affecting all water

---

[1] Currency as of February 2025: 1 USD = 10.75 SEK

consumers (23 factors), forming the conceptual framework for the present study. These identified vulnerability factors are related to adaptive capacity, coping capacity or susceptibility and can be broadly subcategorized into ten categories based on their overall attributes (Table 1), with the full list of vulnerability factors and their corresponding subcategories available in the supplementary materials (S1, S2).

**Table 1. Overview of the number of sectoral, societal, and governance vulnerability factors included in the survey, divided into ten subcategories based on their general attributes. Short names for each subcategory, which are used in the results section, are shown in parenthesis. The subcategories are also marked to indicate whether they involve factors related to adaptive capacity (A), coping capacity (C) and/or susceptibility (S) (Stenfors at al. 2024).**

| Subcategory (Short name) | Total | Sectors | Society | Governance | A | C | S |
|---|---|---|---|---|---|---|---|
| *Societal properties and demographics (Demographics)* | 3 | | 3 | | | | ● |
| *Available funds and financial capacity (Funds)* | 5 | 5 | | | | ● | ● |
| *Characteristics of authority (Authority)* | 6 | | | 6 | | ● | |
| *Presence of irrigation (Irrigation)* | 4 | 4 | | | | ● | |
| *Presence of policies and plans (Policies)* | 9 | | | 9 | ● | ● | |
| *Conditions of surrounding setting (Setting)* | 14 | 9 | 5 | | | ● | ● |
| *Species characteristics (Species)* | 9 | 9 | | | ● | | ● |
| *Presence of Anthropogenic stress (Stress)* | 8 | 8 | | | | | ● |
| *Water supply (Supply)* | 11 | 10 | 1 | | ● | ● | ● |
| *Availability of tools and resources (tools)* | 14 | 6 | | 8 | ● | ● | |
| **Total** | **83** | **51** | **9** | **23** | | | |

## 2.3 Data Collection & Analysis

### 2.3.1 Survey design

In order to find vulnerability factors relevant for Swedish water-dependent sectors as well as society, an online survey was designed based on the vulnerability factors identified by Stenfors et al. (2024). The survey targeted six water dependent sectors: energy (i.e., hydropower, nuclear, thermal etc.), agricultural (i.e., crop, animal husbandry etc.), environmental (aquatic and terrestrial ecosystems), water supply (drinking water production and distribution), water resources (water resource management), forestry (conservation and production) and water intensive industry (i.e. paper and pulp, chemical production, or steel and metal works).

The survey was designed using a 5-point rating scale where stakeholders from different sectors (from now on called "respondents") were asked to rate vulnerability factors based on their perceived impact on drought risk in their sector on a scale from 0 (no impact) to 4 (high impact). The respondents could also opt out by selecting "I don't know" to each factor. After rating the chosen factors, the respondents were asked to rank how confident they were in their rating on a 5-point scale ranging from 0 (highly unsure) to 4 (highly confident).

The survey was divided into three sections: (1) collection of background information on respondents, (2) rating of vulnerability factors for particular sectors, and (3) rating of vulnerability factors for society as a whole. In order to analyze the results in relation to experience, sector, organization type and location, respondents were asked in section one to provide information on their primary sectorial focus, the type of organization they belonged to, and the Swedish county they primarily operated in. They were also asked to judge their level of experience and knowledge concerning drought-related issues in their field on a scale of 0 (no experience) to 4 (large experience). Section two focused on sector-specific vulnerability, where respondents were asked to rate 51 sectoral drought vulnerability factors as well as the 23 vulnerability factors related to governance, on the impact on drought risk in

their sectors. All respondents were presented the same list of sectoral and governance vulnerability factors, regardless of what sector they primarily worked with. This allowed for comparative analysis of what factors are regarded as relevant and irrelevant for the different sectors. Section three addressed societal vulnerability and included the 23 governance factors as well as 9 societal factors, common to all respondents. In this section, respondents were asked to rate the impact of these factors on drought risk for Swedish society. The respondents were also given the opportunity to suggest their own additional factors in both section two and three. As described above, the same 23 governance factors were included in both section two and section three in order to analyze the impact of these factors on sectoral as well as societal drought vulnerability separately.

The survey was designed in English and Swedish, and respondents could choose their preferred language. All survey questions and corresponding response options are presented in the Supplementary materials (S8).

### 2.3.2 Survey recipient selection

With the aim of identifying and selecting potential recipients of the survey, a recipient identification matrix was designed. The matrix used five criteria:

- **Knowledge** – can the recipient potentially provide insight into drought vulnerability in their sector?
- **Sector** – is the potential recipient part of one of our surveyed sectors?
- **Location** – where in Sweden is the potential recipient located and do they improve the geographical spread of knowledge attainment?
- **Organisation type** – does the potential recipient belong to one of our surveyed interest groups (i.e. governmental/local authority, academia/research institute, private/state/municipal owned organization, regional/national sector association, or NGO)?
- **Scale -** does the potential recipient primarily operate on national/regional/local scale (applied to the stakeholder groups: governmental/local authority, NGO and regional/national sector association)?

Using the identification matrix, survey recipients from governmental/local authorities (354 recipients), private/state/municipal owned enterprises (81), academia/research institutes (46), regional/national trade associations (45), and NGOs (35) were identified. Generic contact lists for municipalities (290), counties (21) and authority owned competence centers (31) were collected from official sites (SKL.se and naturvardsverket.se). Contact information for individual recipients within the different organization types were identified using three approaches, (1) internet searches combining their organization name and keyworks in Swedish such as "drought" and "water shortage" (30 respondents), (2) searching their organization websites for keywords "drought" and "water shortage" (37), (3) using the general contact information provided on the organization website (143). Furthermore, individual recipients were found through snowballing, where authors or contributors in articles or projects related to drought or water shortages were included (4 recipients).

### 2.3.3  Analysis of survey responses

All survey responses were transferred and analyzed using Microsoft Excel and RStudio. The survey responses were evaluated using a four-step approach:

I. Data cleaning: Initially, survey responses were screened for respondents answering "I don't know" consistently on all factors. These respondents were removed from further analysis.

II. Data preparation: To analyze geographical differences, responses were categorized as belonging to counties located in northern (above 60 °N) and southern (below 60°N) Sweden.

III. Identification of key factors: Following Meza et al. (2019),  factors were considered relevant for a sector if 50% or more of the respondents within that sector considered them as having medium high or high impact (corresponding to median scores of 3 or 4) on drought risk. For even number responses, the lower integer median was used. Relevant sectoral vulnerability factors were identified using the median rating for each factor, grouped by sector. As the survey only received one response for the sector "Water intensive industry" and one response that did not indicate its sectorial focus, these two respondents were excluded in the analysis of sectorial factors. For societal factors, relevant vulnerability factors were identified using the median rating per vulnerability factor grouped by the respondents' organization type. To handle "I don't know" responses for individual factors, we used a deletion-based available-case method, also known as pair-wise deletion (Xu et al., 2022). This approach excluded respondents who chose the "I don't know" option only from the analyses related to that specific factor. Consequently, we were able to utilize more of the collected data across various analyses, though each factor rating may be derived from a different subset of respondents.

IV. Ranking of key factors using impact scores: The ratings for the factors identified as relevant in step III (ranging from 0 to 4), were normalized to bring them into the range between 0 (no impact) and 1 (high impact), using 0.25 step increments. The factor impact score was then calculated as the mean rating, based on the normalized ratings. Factor impact scores were calculated for each sector and organization type respectively. Factors with an impact score close to 1 are highly impactful on drought risk, whereas indicators with an impact score closer to 0 have less overall impact on drought risk albeit still being relevant for the respondents. The identified impact can be either positive or negative, depending on the vulnerability factor.

To test our hypotheses related to the variability of impact ratings of vulnerability factors (ordinal data) identified as relevant in step III, we utilized the Kruskall-Wallis test by ranks. If significant differences between groups/categories were identified, the data was further analysed using pairwise Wilcoxon rank sum tests to calculate pairwise comparisons between group levels with corrections for multiple testing. Kruskal-Wallis test by ranks and Wilcoxon rank sum, are non-parametric tests that can be applied for ordinal data, and have previously been used for assessing differences between group ratings in studies involving participatory approaches and ordinal data (i.e., Cuesta et al., 2022; Mızrak and Aslan, 2020; Teutschbein et al., 2023a). Hypothesis testing of ratings depending on geographical location and drought experience was only carried out for respondent groups with three or more responses for each response alternative (Geographical location: North versus South, Drought experience: limited (rating 0-1), moderate (2), significant (3-4)). Consequently, differences in factor ratings per geographical location was only carried out for the environmental and forestry sector, and for respondents working in authorities or enterprises. Impact of drought experience on factor ratings was studied for the environmental, water resources and water supply sector and for respondents working in authorities.

## 3 Synthesis of Results

### 3.1 Respondent characteristics and experience

The survey received 108 responses, corresponding to a 19.3% response rate. Six respondents were solely answering "I don't know" and were removed from further analysis. Out of the remaining 102 responses, 61% of respondents were working at an authority (i.e., governmental, municipal, county administrative board) (Table 2). Approximately 19% of respondent were working with research (i.e., in academia or at a research institute), followed by enterprises (private, municipal-, or state-owned) (12%), trade associations (7%), or NGOs (2%). Most of the respondents had a sectorial focus on the environmental (34%) or water supply sectors (15.7%), followed by the water resource (14.7%), forestry (13.7%), agricultural (12.8%), and energy sector (6.9%). Only one respondent was working in a water intensive industry and one respondent did not provide a sectorial focus. Within the sectors, a majority of agricultural respondents were working with crop production (54%), animal husbandry (15%), a combination of crop production, animal husbandry and vegetable production (8%), and the rest reported focusing on other forms of agricultural activities (23%). Respondents from the energy sector were mainly working with hydropower (86%), and all water supply respondents were working with drinking water production and distribution (100%). Roughly half of the respondents from the environmental sector worked with both aquatic and terrestrial ecosystems (46%), or either aquatic (26%) or terrestrial (29%) ecosystems respectively. Most respondents from the forestry sector reported working with forestry production (43%) or nature conservation (29%). The water resources sector mainly consisted of respondents working with water resources management (73%). The majority of respondents (84%) was located in southern Sweden. Respondents from northern Sweden (16%) worked at either an authority, enterprise or with research. Apart from water intensive industries and water resources, at least one response was given for both northern and southern Sweden for all sectors. However, only the environmental and forestry sector received more than two responses by respondents located in northern Sweden. A detailed overview of the respondents can be found in the supplementary materials (S4, S5).

**Table 2. Overview of respondents, their geographical location divided by north (above 60 °N) and south (below 60 °N), and (a) type of organization, (b) primary sectoral focus.**

| | North | South | Total |
|---|---|---|---|
| **a) Type of organization** | | | |
| Authority | 9 | 53 | **62** |
| *Governmental authority* | | *12* | *12* |
| *County administrative board* | *1* | *1* | *2* |
| *Region* | *1* | *4* | *5* |
| *Municipality* | *7* | *35* | *42* |
| *Unspecified* | | *1* | *1* |
| Research | 2 | 17 | **19** |
| NGO | | 2 | **2** |
| Enterprise | 5 | 7 | **12** |
| Trade association | | 7 | **7** |
| **Grand Total** | **16** | **86** | **102** |
| **b) Sectoral focus** | | | |
| Agricultural | 1 | 12 | **13** |
| Energy | 2 | 5 | **7** |
| Environmental | 7 | 28 | **35** |
| Forestry | 4 | 10 | **14** |
| Unspecified | | 1 | **1** |
| Water intensive industry | | 1 | **1** |
| Water resources | | 15 | **15** |
| Water supply | 2 | 14 | **16** |
| **Grand Total** | **16** | **86** | **102** |

More than half of the respondents had more than 10 years of experience in their field of work (59%), and significant experience (experience rating of three or higher) concerning drought-related issues (56%). Respondents from the forestry, agricultural and energy sector had a large share of respondents with significant experience in droughts (over 70% of respondents in each sector) (Table 3). The environmental sector had the largest spread in drought experience, where 37 % indicated having a significant experience of drought. Looking at drought experience by place of employment - enterprise, trade association, or research respondent groups had the highest percentage of respondents with significant drought experience. Respondents from authorities had the largest spread, where 47% of respondents had significant experience of drought. Most respondents indicated that they were moderately confident in the factor ratings they provided for drought vulnerability in their sector (43%) and for society as a whole (47%). Approximately one third of the respondents reported having high confidence in their vulnerability factor ratings concerning vulnerability in their sector (33%) as well as for society as a whole (28%), with the rest reporting little to no confidence in their ratings.

**Table 3. Drought experience as indicated by the respondents by sector as well as place of employment (0-1 signifies little to no experience, 2 moderate experience, 3-4 significant experience of drought-related issues). Respondents from water intensive industries (1) and with unspecified (1) sectoral focus are excluded from the sector count, and only included in the organization counts.**

| Sector/organization | Drought experience (% of respondents) | | | Number of respondents |
|---|---|---|---|---|
| | Limited (rating 0-1) | Moderate (2) | Significant (3-4) | |
| Agricultural | 8 | 15 | 77 | 13 |
| Energy | 14 | 14 | 71 | 7 |
| Environmental | 23 | 40 | 37 | 35 |
| Forestry | | 21 | 79 | 14 |
| Water resources | 20 | 27 | 53 | 15 |
| Water supply | 19 | 19 | 63 | 16 |
| **Grand Total** | **16** | **27** | **57** | **100** |
| Authority | 23 | 31 | 47 | 62 |
| Enterprise | | 33 | 67 | 12 |
| NGO | 50 | | 50 | 2 |
| Research | 5 | 26 | 68 | 19 |
| Trade association | 14 | | 86 | 7 |
| **Grand Total** | **17** | **27** | **56** | **102** |

### 3.2 Relevance of vulnerability factors for sectors, society, and governance

Respondents representing the agricultural sector indicated the highest number of *sectoral* factors as being relevant (i.e. having a median rating of three or higher) on drought risk in their sector (35, out of which 21 with a median rating of four), followed by environmental (32, 4), water resources (31, 12), water supply (26, 9), energy (11, 2), and forestry (10, 0) (Table 4).

**Table 4. The number of (a) sectoral, (b) societal, and (c) governance vulnerability factors included in the survey, the total number of factors considered relevant by one or more water dependent sector or societal organization (i.e., with a median score of 3 or higher), and the number of factors considered relevant by each water dependent sector or societal organization separately. The factor count is divided to represent adaptive capacity, coping capacity and susceptibility as well as subcategories describing the overall attributes of the factors. The subcategories are (see also Table 1): the societal properties and demographics (short: demographics), available funds and financial capacity (funds), characteristics of authority (authority), presence of irrigation (irrigation), presence of policies and plans (policies), the conditions of the surrounding setting (setting), species characteristics (species), presence of anthropogenic stress (stress), available water supply (supply), and availability of tools and resources (tools). As the respondents were asked to rate factors related to governance (c) both from a sectoral as well as societal perspective, factor relevance is included for both sectors and organizations.**

a) *Sectoral vulnerability factors*

| | | Literature review | Relevant Sector (for >=1 sector) | Relevant Society (for >=1 org.) | Sectors | | | | | | Organizations (society) | | | |
|---|---|---|---|---|---|---|---|---|---|---|---|---|---|---|
| | | | | | Agricultural | Energy | Environmental | Water resources | Water supply | Forestry | Authority | Enterprise | Research | Trade association |
| **Sectoral** | **Adaptive** | **10** | **9** | **-** | **7** | **1** | **8** | **7** | **4** | **2** | **-** | **-** | **-** | **-** |
| | Species | 2 | 2 | - | 1 | | 2 | | | 1 | - | - | - | - |
| | Supply | 2 | 2 | - | 2 | 1 | 2 | 2 | 1 | | - | - | - | - |
| | Tools | 6 | 5 | - | 4 | | 4 | 5 | 3 | 1 | - | - | - | - |
| | **Coping** | **10** | **9** | **-** | **8** | **2** | **5** | **6** | **7** | | **-** | **-** | **-** | **-** |
| | Funds | 3 | 2 | - | 2 | | 1 | 1 | 1 | | - | - | - | - |
| | Irrigation | 4 | 4 | - | 4 | | 1 | 3 | 3 | | - | - | - | - |
| | Supply | 3 | 3 | - | 2 | 2 | 3 | 2 | 3 | | - | - | - | - |
| | **Susceptibility** | **31** | **28** | **-** | **20** | **8** | **19** | **18** | **15** | **8** | **-** | **-** | **-** | **-** |
| | Funds | 2 | 2 | - | 2 | | | | 2 | | - | - | - | - |
| | Setting | 9 | 9 | - | 7 | 4 | 6 | 7 | 5 | 4 | - | - | - | - |
| | Species | 7 | 6 | - | 4 | | 3 | 3 | | 4 | - | - | - | - |
| | Stress | 8 | 7 | - | 4 | 3 | 7 | 5 | 5 | | - | - | - | - |
| | Supply | 5 | 4 | - | 3 | 1 | 3 | 3 | 3 | | - | - | - | - |
| | Total | **51** | **46** | **-** | **35** | **11** | **32** | **31** | **26** | **10** | **-** | **-** | **-** | **-** |

b) *Societal factors*

| | | Literature review | Relevant Sector | Relevant Society | Agricultural | Energy | Environmental | Water resources | Water supply | Forestry | Authority | Enterprise | Research | Trade association |
|---|---|---|---|---|---|---|---|---|---|---|---|---|---|---|
| **Societal** | **Coping** | **1** | **-** | **1** | **-** | **-** | **-** | **-** | **-** | **-** | **1** | | | **1** |
| | Setting | 1 | - | 1 | - | - | - | - | - | - | 1 | | | 1 |
| | **Susceptibility** | **8** | **-** | **4** | **-** | **-** | **-** | **-** | **-** | **-** | **4** | **3** | **2** | **4** |
| | Demographics | 3 | - | | - | - | - | - | - | - | | | | |
| | Setting | 4 | - | 3 | - | - | - | - | - | - | 3 | 2 | 1 | 3 |
| | Supply | 1 | - | 1 | | | | | | | 1 | 1 | 1 | 1 |
| | **Total** | **9** | **-** | **5** | **-** | **-** | **-** | **-** | **-** | **-** | **5** | **3** | **2** | **5** |

c) *Governance factors*

| | | Literature review | Relevant Sector | Relevant Society | Agricultural | Energy | Environmental | Water resources | Water supply | Forestry | Authority | Enterprise | Research | Trade association |
|---|---|---|---|---|---|---|---|---|---|---|---|---|---|---|
| **Governance** | **Adaptive** | **9** | **9** | **9** | **8** | **3** | **6** | **8** | **8** | **1** | **9** | **3** | **7** | **8** |
| | Policies | 2 | 2 | 2 | 1 | 1 | 2 | 1 | 1 | | 2 | | 1 | 1 |
| | Tools | 7 | 7 | 7 | 7 | 2 | 4 | 7 | 7 | 1 | 7 | 3 | 6 | 7 |
| | **Coping** | **14** | **13** | **14** | **12** | **6** | **11** | **12** | **12** | **1** | **14** | **4** | **9** | **11** |
| | Authority | 6 | 5 | 6 | 5 | 1 | 3 | 4 | 4 | 1 | 6 | 1 | 4 | 4 |
| | Policies | 7 | 7 | 7 | 6 | 5 | 7 | 7 | 7 | | 7 | 3 | 4 | 7 |
| | Tools | 1 | 1 | 1 | 1 | | 1 | 1 | 1 | | 1 | | 1 | |
| | **Total** | **23** | **22** | **23** | **20** | **9** | **17** | **20** | **20** | **2** | **23** | **7** | **16** | **19** |

When examining the *sectoral* vulnerability factors based on their connection to adaptive capacity, coping capacity or susceptibility, it was observed that all three categories of vulnerability contain factors that are relevant to one or more water dependents sectors. However, factors relating to the susceptibility of the surrounding settings was the only category of factors that was considered relevant by all sectors. Respondents from all sectors found at least 44% of factors relating to this category, as relevant for drought risk. Similarly, factors relating to the adaptive and

coping capacity for water supply, were considered relevant by all sectors except forestry. Instead, the forestry sector mainly found factors relating to species characteristics affecting susceptibility and adaptive capacity, and tools for adaptive capacity as relevant for their sector. Additionally, forestry was the only sector to not find any factors categorized as relating to coping capacity to be relevant for the sector.

Respondents from authorities and trade associations found the largest number of *societal* vulnerability factors as relevant, rating factors relating to all categories of societal vulnerability, except *demographics*, as relevant for societal drought risk. Interestingly, the subcategory *"demographics"* was not considered relevant for societal drought risk by any of the respondent groups. In contrast, almost all vulnerability factors connected to *governance* were relevant for both sectoral vulnerability in at least one sector as well as for vulnerability of society as a whole. Among the sectors, the agricultural, water supply and water resources sectors found the largest number of governance factors as relevant for the sector. When looking at governance factors by place of employment, respondents from authorities found all governance factors relevant for drought risk in society.

### 3.3 Impact scores for vulnerability factors for sectors, society and governance

The evaluation of *sectoral* vulnerability factors revealed that the agricultural sector accounted for several of the highest impact scores, with impact scores for factors concerning irrigation close to 1 (Figure 1). Conversely, the lowest impact scores for the sectoral factors were provided by the energy and forestry sector. The forestry sector tended to rate factors relating to water supply low, giving these factors the lowest impact scores among the sectors. The smallest spread among the impact scores were connected to the conditions of the surrounding settings, where many sectoral factors included in the subcategory received overall medium high to high impact scores by all sectors, even if the factors were not considered relevant for all sectors. Another category of factors that generally see a slightly smaller spread across different sectors is relating to the presence of tools and resources for adaptive capacity. Even though the forestry and energy sector only found a limited number of factors in the category relevant, this category was the only category to not receive impact scores lower than 0.33 by any sector for any of the involved factors. Interestingly, the environmental sector, which had the largest number of respondents among the sectors, gives most factors relating to adaptive capacity similar impact scores. Comparatively, the variation in impacts scores by the environmental sector is larger for factors relating to coping capacity and susceptibility.

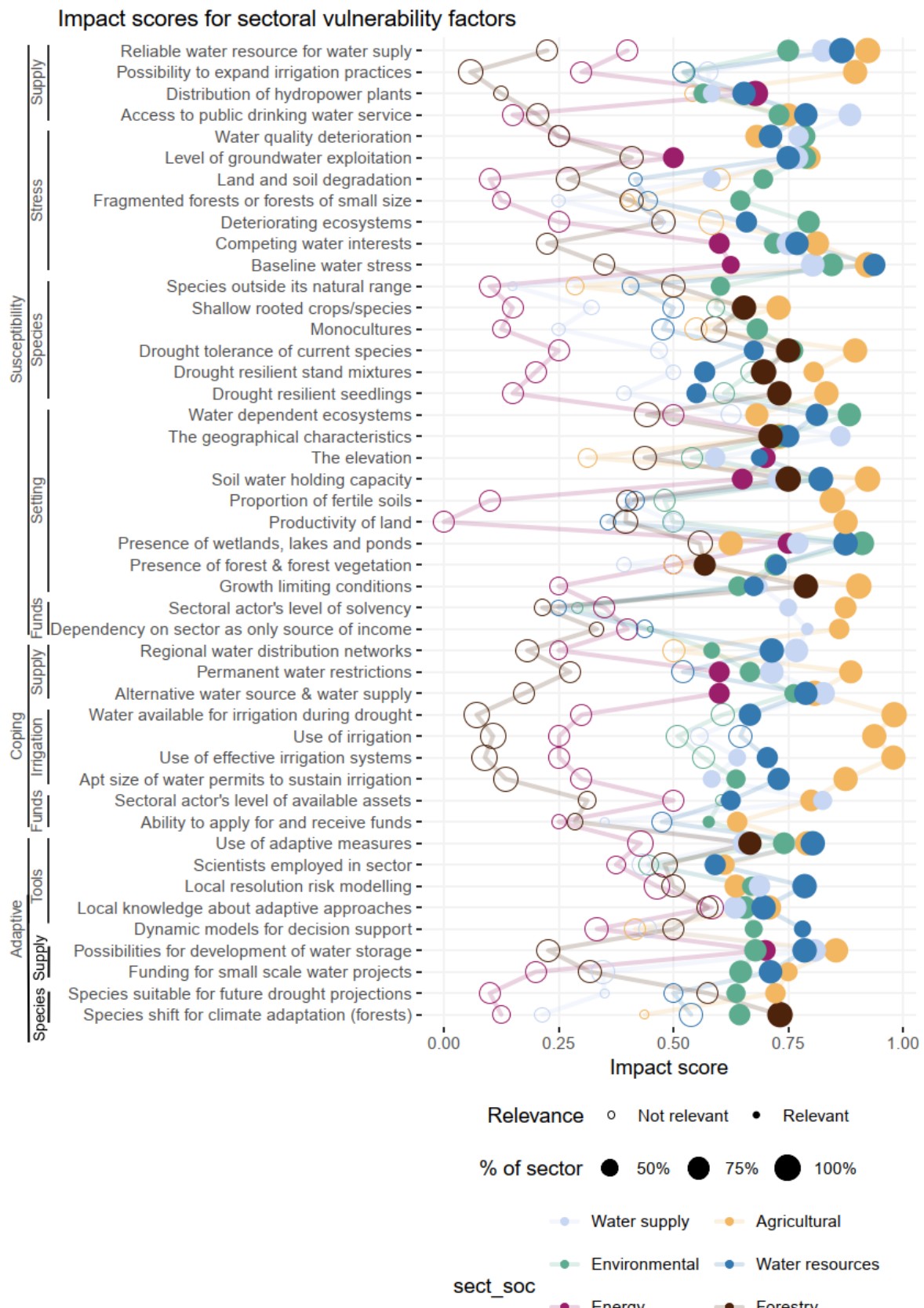

Figure 1. Impact scores for sectoral vulnerability factors concerning adaptive capacity (adaptive), coping capacity (coping), and susceptibility, rated regarding their impact on drought risk in water dependent sectors. Filled dots indicate that the factor is considered relevant for the sector (i.e., with a median score of 3 or higher), whereas open circles indicate that the factor is not considered relevant. The point size signifies the percentage of respondents within a sector that provided an impact rating for the factor.

392

The five relevant factors relating to *society* were all highly rated by respondents from authorities and trade associations, whereas respondents from research, enterprises and NGOs gave slightly lower impact scores (Figure 2). Respondents from all types of organization included in the survey found '*the societal financial dependency on direct water consuming industries (DWC)*' as relevant for societal drought risk. Apart from this, respondents from research only found '*access to public drinking water*' relevant for societal drought risk, whereas NGOs found the '*drought awareness of water users*' relevant as well as highly impactful. Enterprises found '*access to public drinking water*' and '*the size of population*' to be relevant for societal drought risk, but not rating them highly.

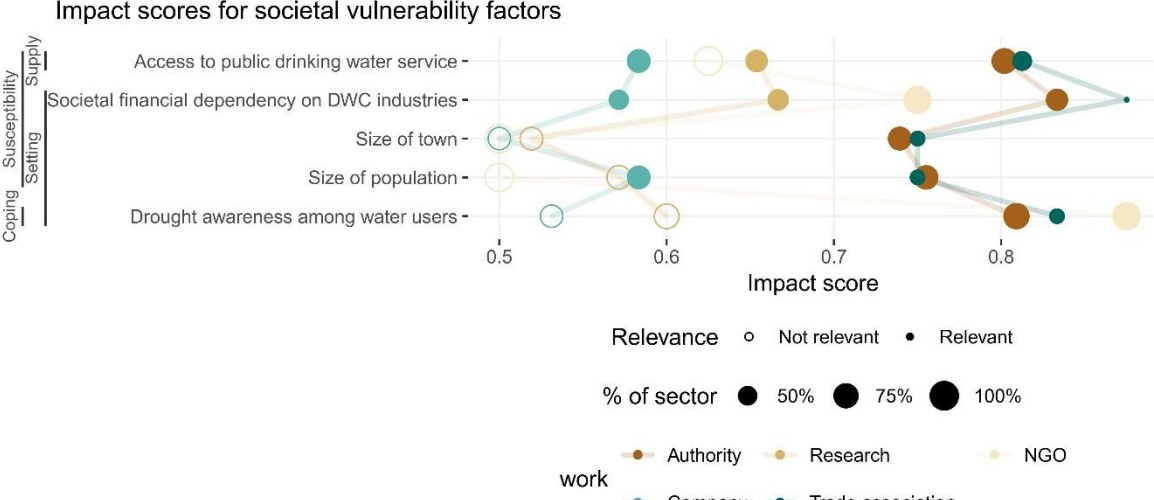

400

**Figure 2. Impact scores for societal vulnerability factors concerning coping capacity (coping), and susceptibility, rated regarding their impact on societal drought risk. Filled dots indicate that the factor is considered relevant by the organization (i.e., with a median score of 3 or higher), whereas open circles indicate that the factor is not considered relevant. The point size signifies the percentage of respondents within an organization that provided an impact rating for the factor.**

Factors relating to *governance* generally receive slightly higher impact scores, both concerning their impact on sectors as well as society, compared to sectoral factors (Figure 3). For example, all factors received impact scores of 0.25 or higher, with the exception for the governance factors '*defined water-use rights*', '*social/physical capacity of authorities to offer drought related support*', and '*building standards relating to water efficiency*'. Apart from two factors relating to having a drought management plan, all factors concerning policies and plans that affect coping capacity, were considered relevant for all sectors except forestry, as well as for society as a whole. Overall, four factors receive impact scores of 0.5 or higher by all sectors: '*drought awareness within authorities*', '*access to relevant data concerning drought*', '*availability of long-term supply and demand assessments*', and '*availability of a drought risk assessment*'. However, no governance factors were considered relevant across all sectors, due partly to the low number of factors considered relevant by the forestry sector. The energy and forestry sector, provided the lowest impact scores for several of the governance factors concerning the impact on drought risk in their sectors. Meanwhile, the highest impact score given by the energy sector, for any sectoral or governance factor, was given to the governance factor '*defined water-use rights*'. This factor was rated highly by respondents across all sectors apart from forestry. Respondents also rated this factor highly when looking at its impact on drought risk for society as a whole. Other governance factors that received high impact scores for sectoral and societal drought risk by at least five sectors were '*having a local water management plan*

or '*an authority-level coordinated water strategy*', '*the drought awareness within authorities*', '*having access to relevant data concerning drought*', and '*long-term supply and demand assessments*'.

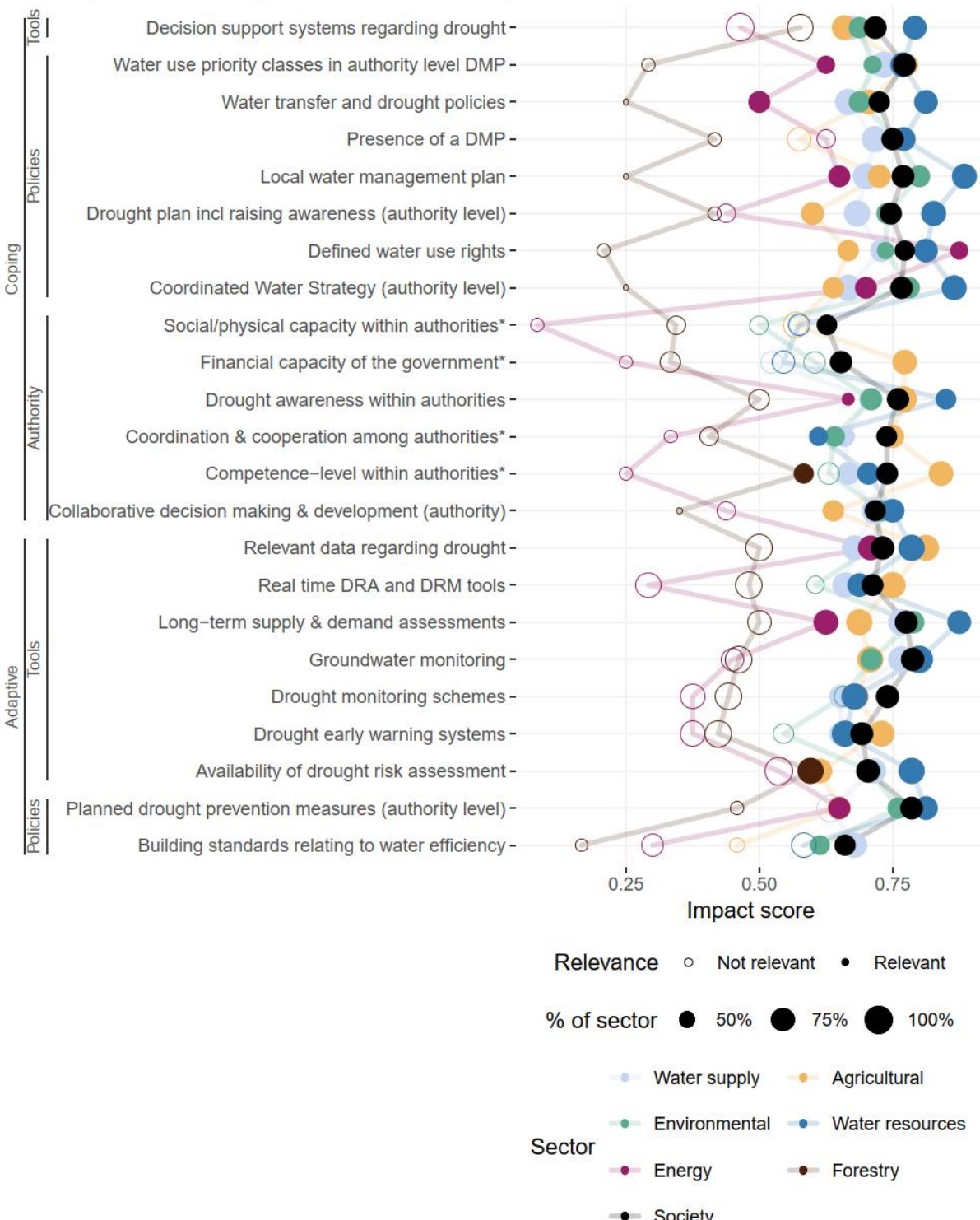

**Figure 3. Impact scores for governance related vulnerability factors concerning adaptive capacity (adaptive) and coping capacity (coping) rated both regarding their impact on drought risk in sectors as well as society as a whole. Filled dots indicate that the factor is considered relevant for the sector or for societal drought risk (i.e., with a median score of 3 or higher), whereas open circles indicate that the factor is not considered relevant. The point size signifies percentage of the respondents within a sector that provided an impact rating for the factor.**

**3.4    Highest rated vulnerability factors for sectors, society and governance**

Looking at the highest rated *sectoral* vulnerability factors across different sectors, most of the factors were connected to coping capacity or susceptibility (Table 5). Only two of the most highly rated sectoral factors concerned adaptive capacity. Certain factors received high impact scores by more than one sector. For example, the soil water holding capacity was among the highest rated factors for the agricultural, environmental and forestry sector. Similarly, the presence of baseline water stress received high impact scores from the agricultural, environmental and water resources sector, and the presence of wetlands lakes and ponds received high impact scores from the energy, environmental and water resources sector. This was reflected in the overall highest rated sectoral factors, for all sectors combined, where the three vulnerability factors received the highest impact scores, followed by the geographical characteristics and having a reliable water resource for water supply.

Adaptive- and susceptibility-related factors were the only vulnerability factors to receive high impact scores by the energy and forestry sector, where the energy sector found the presence of wetlands, lakes and ponds, the geographical characteristics, and the possibilities for development of water storage as factors with high impact on drought risk. The forestry sector on the other hand, gave the highest impact score to having growth limiting conditions, followed by the soil water holding capacity and the drought tolerance of current species.

**Table 5. The five highest normalized impact scores, where the letters in parentheses indicate vulnerability factors concerning adaptive capacity (A), coping capacity (C), or susceptibility (S). Highly rated factors for a sector with the same impact scores are listed arbitrarily.**

**All sectors**

| | |
|---|---|
| (S) Soil water holding capacity | 0,80 |
| (S) Baseline water stress | 0,79 |
| (S) Presence of wetlands lakes and ponds | 0,78 |
| (S) The geographical characteristics | 0,75 |
| (S) Reliable water resource for water supply | 0,73 |

| **Agricultural** | | **Energy** | |
|---|---|---|---|
| (C) Water available for irrigation during drought | 0.98 | (S) Presence of wetlands, lakes and ponds | 0,75 |
| (C) Use of effective irrigation systems | 0.98 | (S) The geographical characteristics | 0,75 |
| (C) Use of irrigation | 0.94 | (A) Possibilities for development of water storage | 0,70 |
| (S) Soil water holding capacity | 0.92 | (S) The elevation | 0,70 |
| (S) Reliable water resource for water supply | 0.92 | (S) Distribution of hydropower plants | 0,68 |
| (S) Baseline water stress | 0.92 | | |

| **Environmental** | | **Forestry** | |
|---|---|---|---|
| (S) Presence of wetlands, lakes and ponds | 0,91 | (S) Growth limiting conditions | 0.79 |
| (S) Water dependent ecosystems | 0,88 | (S) Soil water holding capacity | 0.75 |
| (S) Baseline water stress | 0,84 | (S) Drought tolerance of current species | 0.75 |
| (S) Soil water holding capacity | 0,82 | (A) Species shift for climate adaptation | 0.73 |
| (S) Deteriorating ecosystems | 0,79 | (S) Drought resilient seedlings | 0.73 |

| **Water resources** | | **Water supply** | |
|---|---|---|---|
| (S) Baseline water stress | 0,94 | (S) Access to public drinking water service | 0,88 |
| (S) Presence of wetlands, lakes and ponds | 0,88 | (S) The geographical characteristics | 0,86 |
| (S) Reliable water resource for water supply | 0,87 | (S) Reliable water resource for water supply | 0,83 |
| (S) Soil water holding capacity | 0,82 | (C) Alternative water source & water supply | 0,83 |
| (S) Water dependent ecosystems | 0,81 | (C) Sectoral actor's level of available assets | 0,83 |

Five *societal* factors were considered relevant by the respondents, where the highest rated factor, the financial dependency of society on direct water consuming (short: DWC) industries, was connected to susceptibility. This was followed by having access to public drinking water services, the drought awareness among water-users, the population size and the size of town. Similar to the sectoral ratings, access to public drinking water was also highly rated by sectors such as the agricultural, environmental, and water resources sector and being the highest rated factor for the water supply sector.

Looking at the highest rated *governance* factors, a majority of the factors were related to coping capacity (Table
6). Several of the highest rated factors, when respondents rated the impact of governance factors on their sectors,
received high impact scores by more than one sector. For example, factors such as having a local water
management plan, an authority level coordinated water strategy and the drought awareness within authorities were
among the highest rated governance factors for several sectors. The energy and water supply sector both rated
having defined water use rights highly, whereas respondents from the agricultural and forestry sectors rated the
competence level within authorities to offer drought related support highly. Looking instead at the highest rated
governance factors from a societal perspective, presence of groundwater monitoring appears as the highest rated
vulnerability factor, closely followed by having planned drought prevention measures at authority level, access to
long-term supply and demand assessments, defined water-use rights, and having water-use priority classes in
authority level drought management plans (short: DMP).
**Table 6. The five highest impact scores for a) governance factors rated on their impact on sectoral drought risk, b)**
**societal factors, rated on their impact on societal drought risk, c) governance factors rated on their impact on societal**
**drought risk. The letters in parentheses indicate vulnerability factors concerning adaptive capacity (A), coping capacity**
**(C), or susceptibility (S).**

a) Impact scores for governance factors for
sectors

| **Agricultural** | | **Energy** | |
|---|---|---|---|
| (C) Competence-level within authorities* | 0,841 | (C) Defined water-use rights | 0,875 |
| (A) Relevant data regarding drought | 0,813 | (A) Relevant data regarding drought | 0,708 |
| (C) Water use priority classes in authority level DMP | 0,778 | (C) Coordinated Water Strategy (authority level) | 0,700 |
| (C) Financial capacity of the government* | 0,773 | (C) Drought awareness within authorities | 0,667 |
| (C) Drought awareness within authorities | 0,771 | (C) Local water management plan | 0,650 |
| **Environmental** | | **Forestry** | |
| (C) Local water management plan | 0,800 | (A) Availability of drought risk assessment | 0,596 |
| (A) Long-term supply & demand assessments | 0,788 | (C) Competence-level within authorities* | 0,583 |
| (C) Coordinated Water Strategy (authority level) | 0,781 | | |
| (A) Planned drought prevention measures (authority level) | 0,760 | | |
| (C) Presence of a DMP | 0,760 | | |
| **Water resources** | | **Water supply** | |
| (C) Local water management plan | 0,885 | (A) Long-term supply & demand assessments | 0,767 |
| (A) Long-term supply & demand assessments | 0,875 | (A) Groundwater monitoring | 0,767 |
| (C) Coordinated Water Strategy (authority level) | 0,865 | (C) Water use priority classes in authority level DMP | 0,733 |
| (C) Drought awareness within authorities | 0,850 | (C) Defined water-use rights | 0,729 |
| (C) Drought plan incl raising awareness (authority level) | 0,827 | (C) Presence of a DMP | 0,717 |

b) Impact scores for societal factors for society
as a whole

c) Impact scores for governance factors for
society as a whole

| **Societal factors** | | **Governance factors** | |
|---|---|---|---|
| (S) Societal financial dependency on DWC industries | 0,773 | (A) Groundwater monitoring | 0,788 |
| (S) Access to public drinking water service | 0,750 | (A) Planned drought prevention measures (authority level) | 0,786 |
| (C) Drought awareness among water users | 0,747 | (A) Long-term supply & demand assessments | 0,776 |
| (S) Size of population | 0,695 | (C) Defined water-use rights | 0,773 |
| (S) Size of town | 0,668 | (C) Water use priority classes in authority level DMP | 0,772 |

**3.5    Variations in impact ratings based on sectoral focus, place of employment, geographical location or drought experience**

Our assessment of *sectoral* factors uncovered significant differences in factors ratings depending on the respondent's sectoral focus for 31 sectoral factors (Figure 4a). The forestry sector stood out, having significant differences in impact ratings (p-value <0,05) for several sectoral factors compared to respondents from the agricultural (significant differences seen for 21 factors), environmental (20 factors), water supply (18), and water resources (17) sectors. The main differences were seen for factors connected to water supply, irrigation, and anthropogenic stress. Respondents from the energy sector rated 18 factors significantly different from the agricultural sector, where the differences primarily were seen for factors connected to species characteristics, irrigation and available water supply. Significant differences in factor ratings concerning these three subcategories as well as species characteristics were also seen between the water supply and agricultural sector (11). The impact of place of employment on the sectoral ratings was studied for the environmental, water supply and forestry sector. These sectors had multiple respondents working for at least two different organizations. However, no significant differences were seen in ratings, depending on place of employment for the forestry and environmental sector. For the agricultural sector, significant differences in ratings were only seen for the factor '*Geographical characteristics*' (p-value 0.04), between respondents working in research (median rating 2) and trade associations (median rating 4). An overview of the sectoral factors with significant differences in impact ratings can be found in supplementary materials (S6)

Significant differences in sectoral factor ratings depending on geographical location were only found for the vulnerability factor '*Geographical characteristics*' (p-value 0.03), by respondents from the environmental sector located in southern Sweden (n = 28) versus the north (7), with an average impact score of 0.73 and 1.00 respectively. No significant differences were found between sectoral factor ratings depending on the reported drought experience.

No significant difference in *societal* factor ratings were seen per place of employment. When studying factors from a geographical perspective, respondents from northern Sweden found one societal factor relevant for societal drought risk, compared to four factors by respondents from southern Sweden. Respondents from both north and south of Sweden found the societal financial dependency on direct water consuming industries as relevant for societal drought risk. However, no significant differences in societal factor ratings were seen neither between respondents working in enterprises nor authorities in the two locations.  Similarly, no significant differences in societal factor ratings depending on the reported drought experience were seen.

When looking at differences in factor ratings for the 23 *governance* factors from a sectoral perspective, significant differences were seen for eight factors. Here it was clear that respondents from the forestry sector rated governance factors differently compared to the other sectors, as all significant differences involves the sector. Respondents from the water resources and environmental sector gave significantly different ratings compared to the forestry sector (for seven and five factors respectively), where a majority of the factors were related to water-and drought related policies and plans. When comparing governance factor ratings from a societal perspective per place of employment (Figure 4b), significant differences in factor ratings were found between respondents from authorities and enterprises (2) and research respectively (3). The differences in ratings between the authorities, enterprises and research were seen for the same two governance factors; '*the financial capacity of the government*' and '*the social/physical capacity within authorities*' to offer drought related support, where authorities and research also

rated the presence of water transfer- and drought policies significantly differently. All governance factors that saw significant differences in impact ratings between sectors or organizations are presented in the supplementary materials (S7).

## a) Sectoral vulnerability

| | Water supply 1 | Agricultural 2 | Environmental 3 | Water resources 4 | Energy 5 | Forestry 6 |
|---|---|---|---|---|---|---|
| **Sectoral** | | | | | | |
| **Funds** | | | | | | |
| Dependency on sector as only source of income | | 6 | | | | 2 |
| Sectoral actor's level of available assets | 6 | 6 | | | | 1 2 |
| Sectoral actor's level of solvency | 3 4 5 6 | 3 4 5 6 | 1 2 | 1 2 | 1 2 | 1 2 |
| **Irrigation** | | | | | | |
| Apt size of water permits to sustain irrigation | 2 6 | 1 3 5 6 | 2 6 | 6 | 2 | 1 2 3 4 |
| Use of effective irrigation systems | 2 6 | 1 3 4 5 6 | 2 6 | 2 6 | 2 | 1 2 3 4 |
| Use of irrigation | 2 6 | 1 3 4 5 6 | 2 6 | 2 5 6 | 2 4 | 1 2 3 4 |
| Water available for irrigation during drought | 2 6 | 1 3 4 5 6 | 2 6 | 2 6 | 2 | 1 2 3 4 |
| **Setting** | | | | | | |
| Presence of wetlands, lakes and ponds | | 3 | 2 6 | 6 | | 3 4 |
| Productivity of land | 2 5 | 1 3 4 5 6 | 2 5 | 2 | 1 2 3 6 | 2 5 |
| Proportion of fertile soils | 2 | 1 3 4 5 6 | 2 5 | 2 | 2 3 | 2 |
| Water dependent ecosystems | 3 | | 1 6 | 6 | | 3 4 |
| **Species** | | | | | | |
| Drought resilient seedlings | 2 | 1 5 | 5 | | 2 3 6 | 5 |
| Drought resilient stand mixtures | | 5 | 5 | | 2 3 6 | 5 |
| Drought tolerance of current species | 2 | 1 5 | 5 | | 2 3 6 | 5 |
| Shallow rooted crops/species | 2 | 1 5 | | | 2 6 | 5 |
| Species shift for climate adaptation (forests) | 6 | | | | 6 | 1 5 |
| Species suitable for future drought projections | | 5 | 5 | | 2 3 | |
| **Stress** | | | | | | |
| Baseline water stress | 6 | 5 6 | 6 | 6 | 2 | 1 2 3 4 |
| Competing water interests | 6 | 6 | 6 | 6 | | 1 2 3 4 |
| Deteriorating ecosystems | 3 | | 1 5 6 | | 3 | 3 |
| Land and soil degradation | 5 | 5 | 5 6 | | 1 2 3 | 3 |
| Level of groundwater exploitation | 6 | 6 | 6 | 6 | | 1 2 3 4 |
| Water quality deterioration | 5 6 | 6 | 5 6 | 6 | 1 3 | 1 2 3 4 |
| **Supply** | | | | | | |
| Access to public drinking water service | 5 6 | 5 6 | 5 6 | 5 6 | 1 2 3 4 | 1 2 3 4 |
| Alternative water source & water supply | 6 | 6 | 6 | 6 | | 1 2 3 4 |
| Funding for small scale water projects | 2 3 4 | 1 5 6 | 1 5 6 | 1 5 6 | 2 3 4 | 2 3 4 |
| Permanent water restrictions | 6 | 3 4 6 | 2 6 | 2 | | 1 2 3 |
| Possibilities for development of water storage | 6 | 6 | 6 | 6 | | 1 2 3 4 |
| Possibility to expand irrigation practices | 2 6 | 1 3 4 5 6 | 2 6 | 2 6 | 2 | 1 2 3 4 |
| Regional water distribution networks | 6 | 6 | 6 | 6 | | 1 2 3 4 |
| Reliable water resource for water suply | 5 6 | 5 6 | 6 | 5 6 | 1 2 4 | 1 2 3 4 |
| **Governance** | | | | | | |
| **Policies** | | | | | | |
| Coordinated Water Strategy (authority level) | | | 6 | 6 | | 3 4 |
| Defined water use rights | 6 | 6 | 6 | 6 | 6 | 1 2 3 4 5 |
| Drought plan incl raising awareness (authority level) | | | | 6 | | 4 |
| Local water management plan | | | 6 | 6 | | 3 4 |
| Water transfer and drought policies | | | 6 | 6 | | 3 4 |
| **Tools** | | | | | | |
| Groundwater monitoring | 6 | | | 6 | | 1 4 |
| Long-term supply & demand assessments | | | 6 | 6 | | 3 4 |
| Relevant data regarding drought | | 6 | | | | 2 |

## b) Societal vulnerability

| | Authority 1 | Research 2 | NGO 3 | Enterprise 4 | Trade association 5 |
|---|---|---|---|---|---|
| **Governance** | | | | | |
| **Authority** | | | | | |
| Financial capacity of the government* | 2 4 | 1 | | 1 | |
| Social/physical capacity within authorities* | 2 4 | 1 | | 1 | |
| **Policies** | | | | | |
| Water transfer and drought policies | 2 | 1 | | | |

**Figure 4. Factors where significant differences (p < 0.05) based on pairwise Wilcoxon rank sum test were observed. Differences in vulnerability factors ratings were observed for a) sectoral actors rating sectoral and governance factors,**

**and b) respondents working in different types of organizations rating governance factors for societal vulnerability. One or more coloured and numbered dots in a column indicate that the sector/organization had significant differences in factor ratings compared to these sectors/organizations. The calculated p-values are found in Supplementary material S6 and S7.**

All governance factors were seen as relevant for societal drought risk by respondents from southern Sweden, whereas respondents located in the north found 20 factors relevant. However, no significant differences in governance factor ratings were seen between respondents located in the north, versus south of Sweden, neither for ratings for individual sectors nor society. Significant differences in factor ratings depending on reported drought experience was only seen for the governance factor '*presence of groundwater monitoring*' (p-value 0.042). The factor was generally rated impactful by respondents regardless of the level of drought experience. However, respondents that reported having moderate to moderately significant drought experience (indicating a drought experience rating of two or three), seemed to have a larger spread in their ratings for that factor.

### 3.6    New vulnerability factors identified in the survey

The possibility for the respondents to add their own factors, produced a list of additional sectorial factors (Figure 5). Out of the additional factors added, the only factor mentioned by more than one respondent was "forest fires" which was mentioned by two respondents.

The largest number of additional sectorial factors came from respondents representing the environmental sector, such as factors concerning knowledge of water management among decision makers as well as landowners, and anthropogenic changes to surface waters and water courses (by lowering lake surfaces, dikes, straightening and clearing of water courses). The respondents from this sector also added factors concerning forests, revolving around the area used for production forest and natural forests and presence of forest damages. For the agricultural sector, several of the additional factors suggested revolved around the presence or availability of information on adaptation strategies and knowledge-/evidence-based policies. Financial factors were also mentioned, such as the profitability of investments.

Societal factors included a combination of biophysical and socio-economic factors. For example, forest fires, geography and presence of dikes and other anthropogenic changes to water courses were mentioned as impactful vulnerability factors for Swedish society. From a governance perspective, factors such as planning for climate adaptation and coordination between climate adaptation and civil defense were mentioned. One respondent raised the importance of understanding the actual responsibility of different authority levels during drought.

**Agriculture**
**Coping/Adaptive Capacity**
Knowledge-/evidence based policies
Profitability of investments
Level of competence
Information on adaptation strategies
Mainstreaming adaptation strategies
Farmer's decision-making ability -
under uncertainty
**Hazard/Exposure**
Temperature

**Ecosystems**
**Coping/Adaptive Capacity**
Landowners knowledge in water management
Decision makers knowledge in water mangagement

**Susceptibility**
Presence of forest damages
Presence of pricing of water (withdrawals & use)
Presence of water surface change
Ground slope
Age of permits for on-going water-related activities
Presence of straightened/cleared water courses
Presence of dikes
Ratio between production forest to natural forests

**Water Supply**
**Coping/Adaptive Capacity**
Political competence
Availability of financial models

**Forestry**
**Coping Capacity**
Availability of irrigation of timber storage
**Susceptibility**
Edge effects (i.e. from clear cuts etc.)
**Hazard/Exposure**
Forest fires

**Society**
**Coping/Adaptive Capacity**
The resourcefulness of authorities to implement measures for water supply
during drought (other than available infrastructure for drinking water)
Presence of planning and implementation of climate adaptations
Lack of understanding of responsibility among authorities
Lack of coordination between civil defence with climate adaptation

**Susceptibility**
The geography
Presence of dikes, straightened and/or cleared water courses
Industrial use of public drinking water services,
Industrial use of common surface- & groundwater resources
**Hazard/Exposure**
Forest fires

**Figure 5. New factors that were mentioned by the respondents divided by sector (as expressed by respondents from the agricultural (yellow), environmental (light green), forestry (dark green), and water supply (blue)) sector and societal factors (pink).**

## 4    Discussion

This study investigated societal, sectoral, and cross-sectoral drought vulnerability factors in a Nordic country based on stakeholder perceptions. Respondents from authorities, private enterprises, research, trade associations and NGO from seven water-dependent sectors were given the opportunity to rate the impact of numerous vulnerability factors, some of which had not been previously used in drought vulnerability assessments for their sector, providing new insights into sectoral and societal drought vulnerability.  Results show that each sector has their unique vulnerability profile, however some vulnerability factors are impactful for more than one sector. Among the included sectors, the forestry sector stands out, finding only a low number of listed vulnerability factors to be relevant.

### 4.1    Capturing the complexity of drought vulnerability

The fact that approximately 90% of the factors used in the literature were deemed relevant by the stakeholders underscores the broad range of elements that contribute to sector-specific and societal vulnerabilities. The results highlight the complexity of drought vulnerability, showing that a combination of several different factors impacts the overall drought risk of a sector or society as a whole. This finding aligns with studies from other world regions. For instance, Moshir Panahi et al. (2023) found that 44 vulnerability factors played a role in drought vulnerability for Iran when using an impact-based method combined with expert weighting, arguing that the wide range of effective drought vulnerability factors, provides evidence of its complexity. Similar to our results, Moshir Panahi et al. (2023)  found that water resources, ecological, and agricultural systems had some of the largest number of factors driving vulnerability.  For agricultural and ecological systems, this could reflect the sectors' reliance on both the ecological resilience of species and surrounding settings, as well as on access to reliable water supply. Interestingly, the results show that the respondents working with water resource management, also see factors relating to both ecological resilience as well as water supply as important for the sector. This emphasizes the role of terrestrial ecosystems services for water resource management (Tidwell, 2016), as drought impacts on ecosystem health can affect for example water retention, water quality (Stefanski et al., 2025) as well as stream temperatures (Raheem et al., 2019).

In Meza et al. (2019)'s survey on global drought indicators for agriculture and water supply, 45 vulnerability indicators out of 64 total, were rated as relevant for the agricultural sector. When comparing the top five most relevant indicators identified in their study (i.e. dependency on agriculture for livelihood, cultivation of drought resistant crops, irrigated land, existence of adaptation policies & plans, degree of land degradation & desertification), all factors are among the sectoral and governance factors rated as relevant by agricultural respondents in Sweden. However, when comparing their final impact scores, only the presence of irrigation is included in the five highest rated factors in both studies. Somewhat surprisingly, in contrast to the results from Meza et al. (2019), baseline water stress is not among the five highest rated vulnerability for Swedish water suppliers. However, both baseline water stress and competing water interest do receive high impact scores by the respondents working in the water supply sector. Still, these findings highlight the importance of locally-relevant vulnerability factors, and that the relevance of vulnerability factors can differ depending och climatological and ecological context.

This information is crucial, as indicator-based assessments often use the same set of vulnerability indicators, regardless of sectoral, contextual and scale-dependent differences (Hagenlocher et al., 2019). The contextual nature of drought vulnerability becomes clearer when comparing our findings with the most commonly used vulnerability indicators globally. For example, factors related to poverty and income are some of the most commonly used vulnerability factors used in people-centered drought risk assessments globally (Hagenlocher et al., 2019). Yet, none of the factors connected to demographics, which have been described and used in several articles as factors connected to drought vulnerability, such as the level of social integration (Alcamo et al., 2008; Erfurt et al., 2019; Hurlbert and Montana, 2015) and socio-economic susceptibility of the population (Acosta & Galli, 2013; Hurlbert & Gupta, 2017; Pappné Vancsó et al., 2016; Raikes et al., 2021) were considered relevant for societal drought vulnerability by the respondents. This contrasts the findings of Englund et al. (2023), who, through participatory approaches, found that several aspects connected to social susceptibility (e.g., Illness & disability, age, income, unemployment, housing) and social integration where important for social vulnerability to floods in the Swedish municipality of Halmstad. Likewise, Turesson et al. (2024) concluded that social vulnerability to climate extreme exists in Nordic countries, finding factors related to for example demographics, income, and social cohesion as contributing to social vulnerability to climate risk in the study region. However, as these studies are not specifically focused on drought hazard but on flood and climatic hazards in general, our results could be an indication that aspects such as social integration and socio-economic susceptibility are less impactful for drought risk in welfare states such as Sweden. Another potential reason for this disconnect could lie in whether societal vulnerability is addressed as an individual condition or a societal one (Orru et al., 2022). In our survey, respondents were asked to rate the societal vulnerability factors regarding their impact on drought risk in society as a whole. Whilst social integration and socio-economic susceptibility may play an important role for drought risk for the individual, the respondents may view such factors as having less impact on overall societal drought risk. Alternatively, the low rated relevance for these factors, can be a reflection of an unwillingness to categorize certain groups or individuals as more vulnerable than others (Orru et al., 2022), and instead putting an emphasis on structural pressures and institutional tools for societal vulnerability. Looking at the factors considered relevant for societal drought risk, these could be seen as relating more to factors exerting pressures on a society,

such as the size of population, size of town and the societal dependency on water dependent sectors. Nevertheless,
further research is needed to understand the role of social susceptibility for drought vulnerability in the region.

Local differences in stakeholder perceptions of societal vulnerability were also seen. Only one out of the nine
societal vulnerability factors was seen as relevant to societal drought risk by respondents located in the northern
part of Sweden, while respondents from southern Sweden rated four factors as relevant. The reasons for the low
number of relevant societal vulnerability factors in the north of Sweden could be several. For example, southern
Sweden has a much higher population density, as all urban areas with a population over 100 000 inhabitants, are
located below the 60th parallel. Future climate projections also generally indicate a general wetting trend in the
northern parts of Sweden, and drying trends in southern Sweden (Chen et al., 2021; Sjökvist et al., 2019;
Teutschbein et al., 2023b) that could potentially affect drought vulnerability and the perception thereof.
Furthermore, in a study by Teutschbein et al. (2023a) studying drought severity and perceived impacts of the 2017
and 2018 drought years by Swedish municipalities it was shown that the perceived impacts of the drought events
decreased in a poleward direction. The study found that the municipalities located north of the 60th latitude
perceive none or very weak impacts from the two drought years, as compared to municipalities located south of
the 60th latitude who saw differing perceptions of the impacts, ranging from no impacts to very strong impacts. In
addition, southern municipalities experienced on average more severe drought conditions than northern
municipalities during the 2018 drought event (Teutschbein et al., 2023a). Such differences could potentially affect
the overall perception of drought risk and drought vulnerability.

In contrast to social and societal vulnerability, some biophysical factors seem to be less contextual, and could
potentially play a more general role to drought vulnerability regardless of climate and, to some extent, sectoral
context. For example, factors related to soil characteristics, topography, and water resources have commonly
been used globally for drought vulnerability assessments (González Tánago et al., 2016), and were considered
relevant for all or most of the sectors included in this study. Likewise, technical factors such as the use of irrigation
have commonly been included in vulnerability assessments (González Tánago et al., 2016; Hagenlocher et al.,
2019) and were shown to be highly relevant for the agricultural sector by the respondents. Importantly,
respondents from the water supply and water resources sectors also stressed the need for effective irrigation
system, potentially due to the potential strain on water availability that irrigation can cause.
**4.2    New insights to drought vulnerability**
Letting all sectors rate the same list of factors provided new insights into the perceptions of drought vulnerability,
for example for the environmental and energy sector. Stenfors et al. (2024) only found a limited number of factors
concerning drought vulnerability for the environmental and energy sector in the literature. However, respondents
from these sectors found several sectoral and governance factors to be relevant for their sectors. Additionally, the
environmental sector also introduced the largest number of new factors suggested by the respondents, such as
landowners' and decisions makers' knowledge in water management, presence of straightened water courses or
dikes and the age of water permits. This implies that there may have been a significant knowledge gap in how
drought vulnerability has been assessed for these sectors historically, versus how it is perceived in the sectors.
Interestingly, the results in this study also showed that the energy sector shared all of its relevant vulnerability
factors with several other water-dependent sectors, indicating that there are common interest points between the
energy sector, and sectors such as the environmental sector and water resources. The results illustrate the
importance of stakeholder engagement and participatory approaches to understand vulnerability. Local expert
knowledge not only helps in assessing the relative impact and relevance of standard drought vulnerability factors,
but can also introduce new information on how drought vulnerability can be used in future risk assessments(Asare-
Kyei et al., 2015).

The design of the survey also enabled analysis of how vulnerability to drought is perceived within sectors, as well
as across sectors. Whilst all sectors were shown to have unique vulnerability profiles, similarities across sectors
could be found, and all sectors shared factors that were considered relevant and impactful for drought risk with
one or more sectors. The results underscore the systemic nature of risk, where underlying societal and sectoral
drought vulnerabilities can be relevant for more than one sector. They also highlight the relative importance of
these vulnerability factors in a multi-sectoral and interconnected system, thereby adding valuable insights for, for
example, conducting drought risk assessments and designing adaptation policies (Hagenlocher et al., 2023;
Stefanski et al., 2025).    Much like what was described by Stenfors et al. (2024), respondents showed that
vulnerability factors describing drought vulnerability of direct or indirect water consumers, should be combined
with factors concerning the governance process, policies, tools and plans that exist to provide drought related
support. This provides a roadmap for researchers and policy makers conducting drought vulnerability or risk
assessments, indicating that a starting point to any assessment is by identifying the water dependencies present in
the studied systems and the vulnerabilities directly connected to such stakeholders. This should then be combined
with analysis of the institutional arrangements, policies, and other tools and how they function in providing
drought related support in these systems. For example, the results show that governance factors connected to the
presence of water- or drought-related policies were relevant for most sectors, receiving high impact scores from
all sectors, except forestry, as well as for society as a whole. This suggests that policy instruments do play a crucial
role for lowering drought vulnerability in socio-hydrological systems, both on a sectoral and societal level. This
is consistent with several studies on climate risk, disaster and drought management, all arguing that adaptive
governance is essential for managing climate-related risks (e.g., Dias et al., 2022; Hurlbert & Gupta, 2016;
Nelson et al., 2008).  Additionally, the results also show that not only policy instruments are of importance, but
also the characteristics of authorities. For example, a majority of sectors showed the importance of drought
awareness within authorities and the level of competence, coordination, and cooperation of authorities to offer
drought related support. This is in line with Hurlbert and Montana (2015), who stress the importance of
competence for responsiveness in water governance. Whilst such factors are difficult to incorporate in applied
vulnerability assessments, they highlight the importance of competence-building activities, and cooperation and
coordination within and across authorities for managing drought risk. It also shows a limitation of strictly
indicator-based vulnerability assessments. Basing vulnerability assessments on strictly measurable, available data
provides a starting point for assessing vulnerability, but may overlook important aspects of system vulnerability
that cannot be measured or included in such types of assessments. Consequently, the results of indicator-based
vulnerability assessments should preferably be put in a wider perspective, stressing such limitations.

From a sectoral perspective, the results highlight that combining governance factors with factors related to the
surrounding settings, anthropogenic stress (such as baseline water stress, or competing water interests), and water
supply is important when conducting drought vulnerability assessments incorporating vulnerability to drinking
water production, energy production, agriculture and the environment. Such factors should then be combined with
sector specific vulnerabilities, to further enhance the quality and detail of vulnerability assessments. For such
assessments, the calculated impacts scores not only show the impact of the factors within a specific sector, but
highlight the patterns of interconnectedness between these sectors. Out of all the sectoral factors, the only two
factors that were considered relevant for all sectors were the soil water holding capacity and the geographical
characteristics. This indicates that these vulnerability factors are relevant across all water-dependent sectors
included in this study, and should preferably be included in drought vulnerability assessments, if possible. It also
underscores, the issue of using the same set of vulnerability factors independent of sectoral perspectives
(Hagenlocher et al., 2019). Whilst significant differences in factor ratings could be seen between the forestry
sector and all other sectors, both regarding sectoral as well as governance factors, the sector shared most of its
identified factors with at least one sector. Incorporating the forestry sector in holistic risk assessments or nexus-
approaches has been stressed (Melo et al., 2020; Tidwell, 2016), and the results provide a starting point for
inclusion of the sector in factor-based vulnerability assessments. However, whilst both the forestry and
agricultural sector share factors relating to for example species characteristics, it is important to note the
differences in crop/species rotation time between the two systems. Such temporal differences may have an impact
on for example the two sectors' capacity to adapt in the short term. However, as this article mainly focuses on
differences in vulnerability perspectives depending on sectoral, societal and regional differences, further research
is needed to better understand the influence and incorporation of temporal scales in vulnerability assessments and
adaptation efforts.

In the conceptual framework proposed by Stenfors et al. (2024), the sectoral factors for assessing drought
vulnerability in direct water consuming sectors can be divided into vulnerability factors relevant when studying
droughts on blue or green water resources respectively, as well as factors that are universally relevant for all direct
water consuming sectors, regardless of where in the hydrological system the drought is located. In the conceptual
model, blue water entails water available as surface or groundwater, while green water represents water stored as
soil moisture in the unsaturated zone (Falkenmark and Rockström, 2006). Consequently, the most relevant
vulnerability factors for a sector would be related to whether or not they are mainly dependent on blue or green
water resources. This could be seen in some of the sectoral ratings, where respondents from the energy, water
supply and water resources sectors tend to give lower impact scores or find certain factors irrelevant for their
sectors, whilst the same factors receive high impact scores from respondents from the forestry, environmental and
agricultural sector. Conversely, in certain subcategories of sectoral factors, such as available water supply, and
availability of water and/or drought related policies and plans the forestry sector does not find any factor relevant
for their sector. Further research is needed to better understand how the type of water dependency can influence
the relevance of vulnerability factors as well as their impact scores as this would have implications on how factors
are chosen when performing vulnerability assessments.
**4.3    Implications for whole of society adaptation and policy design**
The results provide details on potential areas for adaptation both for specific sectors as well as from a multi-
sectoral perspective. For example, several factors relating to water supply and anthropogenic stresses on water
supply were considered relevant for many of the sectors. This gives an indication that efforts on for example
minimizing water consumption (to reduce baseline water stress), and sourcing reliable or alternative water sources
are valuable tools for addressing drought risk for several sectors. Combining such efforts with tools for
groundwater and drought monitoring, and policy development focusing on water management and drought
prevention is important. Here, respondents stress the importance of having defined water use rights, and water use
priority classes. Efforts should also be focused, not only on policy development, but for competence building
within and across authorities to be able to provide drought support for a range of water dependent-sectors in the
case of droughts, and for coordination and cooperation across authority levels.
## 4.4    Incorporating stakeholder knowledge
Integrated stakeholder engagement in drought risk reduction is important for building drought resilience
(UNDRR, 2019), and bridging the gap between scientific and practical knowledge (Moreira et al., 2023). As
vulnerability is contextual, participatory approaches can provide valuable insights into regional or local
circumstances (Martín et al., 2017; Moreira et al., 2023) and guide effective adaptation research and planning
(Fleming et al., 2023). However, participatory approaches come with uncertainties. For example, it has been
argued that the results of participatory approaches can be affected by the priorities of the involved stakeholders,
as the perceptions of vulnerability will differ between local community members, industries, NGOs or
authorities(Fleming et al., 2023). Our study addressed this by incorporating stakeholder working in authorities,
academia, industry, trade associations and NGOs. Whilst significant differences in sectoral factors ratings were
seen between sectors, the place of employment only generated significant differences in ratings for a limited
number of factors. For sectoral vulnerability, this could be an indication that perception of drought vulnerability
among the respondents is more closely connected to sectoral knowledge than differences in place of work. This
was described by Siegrist and Árvai (2020), who suggested that people working in the same field tend to have
similar risk perceptions due to their shared domain-specific knowledge and expertise. However, for societal
vulnerability, no significant differences were seen depending on the place of employment or the reported drought
experience of the respondents. This could imply that the ratings of societal vulnerability factors are influenced
more by the respondent's subjective opinion, rather than the characteristics of the respondents, such as work or
drought experience (Moreira et al., 2023). However, since a majority of stakeholder were working for authorities,
the overall distribution of place of employment among the respondents may still have an effect on the overall
results. For example, the impact scores of governance factors on societal vulnerability, are similar to those given
for governance factors for the environmental, water supply, and water resources sectors. These respondent groups
were the three largest, with a large percentage of respondents working for authorities.

## 4.5    Further work
Our drought vulnerability survey found several vulnerability factors relevant for drought vulnerability in water-
dependent sectors, and identified new factors that can be used when studying drought vulnerability in forested
cold climates. However, relevant factors for water-dependent industries such as paper and pulp production,
chemical production and steel and metal works could not be explored due to the limited amount of responses
attained. The study provides a comprehensive list of context-specific drought vulnerability factors, as well as their
relative impact on drought risk depending on sector, but more work is needed to operationalize the factors through
suitable indicators. The results are a starting point for exploring drought vulnerability in forested cold-climate
countries (primarily in northern America and north-eastern Europe), and future research should aim to incorporate
the factors in applied assessments to deepen the understanding of drought risk in the region.

**5    Conclusion**

To confirm and investigate relevant vulnerability factors for forested cold climates, respondents from seven water-
dependent sectors employed in five different types of organizations rated drought vulnerability factors based on
their perceived impact on drought risk in their sector and on society as a whole. As hypothesized, impact ratings
differed depending on sectoral focus of the respondents, as well as place of employment for sectoral and societal
vulnerability factors respectively, where significant differences in vulnerability ratings were seen for several of
the studied factors. Furthermore, geographical differences could be seen in the number of societal vulnerability
factors rated as relevant when comparing responses based on respondents' reported geographical location.
Significant differences between ratings made by respondents with little to no experience of droughts compared to
respondents with larger reported experience was only seen for the vulnerability factor '*presence of groundwater*
*monitoring*'.
The conceptual framework proposed by Stenfors et al. (2024) for drought vulnerability in forested cold climate
regions as well as the vulnerability factors it was based on, was further investigated based on the survey results.
Differences in sectoral and governance related vulnerability factor ratings were seen for the included sectors.
Looking at vulnerability for society as a whole, all vulnerability factors related to governance were found relevant,
whereas only five societal factors were seen as relevant to drought risk by the respondents.
As previous drought events have shown, countries located in forested cold-climate zones are not exempt of
drought events. The large list of vulnerability factors, identified as impactful by the sectoral stakeholders in this
study, gives an indication of the complexity of drought vulnerability and the many facets in which it can affect
societal sectors in these regions, ranging from available water supply, to the presence of drought-oriented policies
and plans. However, factors such as the '*soil water holding capacity*' and the '*geographical characteristics*' were
considered relevant by all included sectors and should preferably be included in future sectoral drought
vulnerability assessments in these climates.  As there is a current lack of drought risk and vulnerability assessments
in some forested cold countries such as Sweden, efforts should be made to further analyze the results obtained in
this study for operationalizing the factors through development of relevant drought indicators and identification
of suitable data sources. In this context, our study provides a valuable guide into drought vulnerability for six
water-dependent sectors as well as for society as a whole to effectively lower drought vulnerability in water-
dependent societies.

**6    Data availability**

The data supporting the findings of this study were collected through an online survey with stakeholders, with
assurances provided that the data would be anonymized and used solely for the purposes of the corresponding
author's PhD project. Due to these ethical considerations and the privacy of the respondents, the data cannot be
made publicly available. However, detailed information about the study design, data collection methods, and
analysis procedures are provided within the paper. For any inquiries regarding the data, please contact the
corresponding author.

**7    Author Contribution**

ES: Conceptualization (lead); data curation (lead); formal analysis (lead); investigation (lead); methodology
(lead); project administration (supporting); validation (lead); visualization (lead); writing – original draft (lead);

writing – review and editing (lead). MB: Supervision (supporting); visualization (supporting); writing – review and editing (supporting). TG: Conceptualization (supporting); methodology (supporting); supervision (supporting); visualization (supporting); writing – review and editing (supporting). CT: Conceptualization (supporting); funding acquisition (lead); methodology (supporting); project administration (lead); supervision (lead); visualization (supporting); writing – review and editing (supporting).

**8    Competing interests**

The authors declare that they have no conflict of interest.

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
