# Peer review of "Multi-sectoral and systemic drought risk in forested cold"

_EGUsphere, 2024_

## Author Response (AR1)

Dear Assoc. Professor van Loon

Thank you for your thoughtful and constructive feedback on our manuscript titled "User-validated Drought Vulnerability Factors in Forested Cold Climates: Multi-sectoral Perspectives from Sweden" with ID number egusphere-2024-1988. We greatly appreciate the time and effort you and the reviewers dedicated to assessing our work and providing insightful comments. Your feedback has been invaluable in improving the quality and clarity of our paper.

In the revised manuscript, we have tried to incorporate the valuable suggestions given by the reviewers. We have especially focused on restructuring and revising the introduction to better establish the manuscript's aim and objectives. Additionally, the discussion section has been revised to provide a more in-depth analysis of the results, their implications, and their contribution to existing knowledge.

We have also meticulously rechecked all our results to ensure consistency and validity. During this process, we identified and corrected minor errors, resulting in updates to Tables 4 and 5, and minor corrections in the text. The colored cells in Tables 5 and 6, have also been changed to black font, based on your advisement following the original manuscript submission.

All changes made to the manuscript are specified using the Track Changes feature in Microsoft Word. Detailed point-by-point responses to the reviewers' comments are given below (*in blue italic*), with line numbers referring to the revised manuscript version where all changes have been accepted.

We hope that you and the reviewers find that the revisions have improved our manuscript and that it is now suitable for publication in NHESS Special issue: Drought, society, and ecosystems (NHESS/BG/GC/HESS inter-journal SI).

Best Regards,

Elin Stenfors, PhD student
On behalf of the (co-)author team

**1. Reviewer 1**

My may concern is the novelty of the Study. As I mention in comment 11, in general, although the presentation of the method and the description of research objectives are well crafted and easy to follow, I do not see a significant novelty in the method or in the objectives necessary for the HESS journal. The identification of vulnerability factors that the authors used in their questionnaire is built on a "Review paper" that the authors have published recently (Stenfors et al., 2024) and basically come from other studies. The Review paper is an important contribution to the field, but this current manuscript (User-validated…) does not add much to our understanding of the identified vulnerability factors identified in the Review paper.

*Response: Thank you for your thorough review of our manuscript. We appreciate the effort to provide such a detailed feedback.*

*We appreciate the reviewer's careful assessment and valuable suggestions regarding novelty and methodology. We acknowledge the concern about potential overlap with our earlier review paper. However, we respectfully argue that the current manuscript makes distinct and substantial contributions beyond the scope of the previous review, particularly through its empirical validation and deeper examination of vulnerability factors. While the review paper (Stenfors et al., 2024) synthesized and conceptually organized vulnerability factors from existing literature, it did not empirically validate or test these factors in the geographical context of cold-climate regions. For example, the review highlighted irrigation as a prominent vulnerability factor. However, in Scandinavia, irrigated agriculture accounts for only 3% of all agricultural land, suggesting its limited significance in this context. Our current study directly addresses this gap by empirically validating previously identified factors through a structured questionnaire and stakeholder engagement within the study region. This empirical approach is essential for translating theoretical insights into actionable knowledge, demonstrating how these factors manifest in practice. By bridging this gap, our study enhances the practical relevance and applicability of the earlier theoretical synthesis. We have clarified this in the revised manuscript (e.g. Lines 589-598, 635-643, 646-652, 661-668, 681-684, 687-691).*

Even the area that they have studied here is a part of the study area in the Review paper.

*Response: Although the current study area overlaps geographically with the one discussed in the review, the primary intent here is precisely to empirically test theoretical findings within a real-world context. The objective is not merely repeating the review findings, but rather advancing the knowledge by context-specific testing, offering stakeholders' concrete, actionable insights tailored to local management and decision-making. We have clarified this in the revised manuscript (Lines 67-73, 101-103).*

I believe the authors could have already included the results of this current manuscript in that review paper to show an empirical example of vulnerability factors identified by other studies and avoid salami-slicing their well-presented research.

*Response: We fully recognize and appreciate the concern regarding the risk of "salami-slicing." However, our intention has always been clear: the review paper was purely conceptual and synthesizing in nature, whereas the present manuscript was designed explicitly as the logical next step - an independent, empirical investigation that tests the theoretical concepts outlined previously. The substantial methodological difference and new, empirically derived insights firmly distinguish it as a standalone contribution.*

I noticed the manuscript lacks a section detailing the design and classification process for the survey questions. This study could stand alone as a robust paper if it included a more sophisticated analysis in the questionnaire design phase. For example, the questions could be categorized independently of any prior knowledge of vulnerability factors. Methods such as PCA could then be applied to validate how respondents' answers align with the categories you've identified.

*Response: Thank you for your valuable comment. The manuscript already contained a dedicated section detailing the survey design. While the suggested methodology involving PCA is valid and insightful, our study intentionally applies a multi-stage expert-driven factor identification and refinement approach. Specifically, we started with a comprehensive review encompassing a large number of potential vulnerability factors, then systematically reduced these through rigorous literature synthesis, and subsequently refined this list further through targeted*

*expert input and empirical validation in our study region. This structured, top-down refinement provides a robust empirical verification and prioritization of previously theoretically derived vulnerability factors.*

**◼ My detailed comments and suggestions:**

1. I suggest shortening the title: "Drought Vulnerability Factors in Forested Cold Climates: Multi-Sectoral Insights from Sweden".

   As for the term user-validated, it could be slightly misleading if the validation is based only on survey responses. Typically, user validation implies that users have actively confirmed the factors' relevance or usefulness through direct engagement or testing. If respondents simply provided perspectives, terms like user-informed and stakeholder-informed might better reflect that the factors are shaped by input across various sectors.

   *Response: Thank you for your suggestion. We have revised our title to better reflect our work. We have also changed the phrase "user-validation" to stakeholder "perceptions" (Line 92,552, 618, 633, 645, 753).*

2. Line 21 (Abstract): I recommend removing 'Northern Europe' from the sentence, as the study site is specifically in Sweden. Referring to Northern Europe implies a broader region, which may encompass different socio-hydrological contexts that are not representative of your study area.

   *Response: The term has been removed.*

3. Line 25 (Abstract): I recommend removing the word 'successfully' from the sentence. Identifying vulnerability factors from surveys is not something that implies success or failure. It's more about conducting a thorough analysis rather than achieving a specific outcome. A more neutral phrasing would improve the clarity and objectivity of this statement and throughout your entire manuscript. Also, I believe that using the term "investigation" instead of identification is more suitable for the concept of "vulnerability factors" in this study as you investigated these factors through surveys. In your Discussion, line 531, you have correctly used the term "investigation".

   *Response: The term has been removed. and we have revised the manuscript according to your suggestion (Line 551, 778, 789)*

4. Lines 44-45: When you mention that "there is currently…", I expect to see some current literature supporting your argument. Do we know that since 2013 and 2018 there has not been any consensus to define drought vulnerability?

   *Response: We have added more recent references that discuss the lack of consensus, to support our claim (Line 50-51).*

5. Line 57: I suggest defining cascading effects before giving the examples.

   *Response: We have rephrased the sentence to clarify the term (Line 44).*

6. Line 91: As mentioned in comment #1, I suggest replacing "user-validation" with terms like "user-informed".

   *Response: The term "user-validation" has been replaced with stakeholder "perceptions" (Line 92,552, 618, 633, 645, 753).*

7. Line 128: Typo: Divided?

   *Response: Thank you for identifying a typo. It has been corrected in the revised manuscript. (Line 132)*

8. Line 171: Try to be consistent with the orders throughout the manuscript, i.e., susceptibility, coping, adaptive. This may sound trivial, but it helps the readers to follow the rhythm of your story more easily.

*Response: While we had tried to be consistent with the ordering of our terminology, there were still instances with inconsistent ordering in the manuscript. We thank the reviewer for pointing this out, and have now carefully revised the manuscript for consistency in the ordering (Line 432, 446, 468-469).*

9. Table 1 caption, refer to my previous comment. Also, I suggest changing "based to" to either "according to" or "based on".

   *Response: The manuscript has been revised accordingly (Line 195).*

10. Table S3: Please correct the caption, Societal v factors.

    *Response: The table caption has been revised accordingly.*

11. Method Section: In general, although the presentation of the method and the description of research objectives are well crafted and easy to follow, I do not see a significant novelty in the method or in the objectives necessary for the HESS journal. The identification of vulnerability factors that the authors used in their questionnaire is built on a "Review paper" that the authors have published recently (Stenfors et al., 2024) and basically come from other studies. The Review paper is an important contribution to the field, but this current manuscript (User-validated…) does not add much to our understanding of the identified vulnerability factors identified in the Review paper. Even the area that they have studied here is a part of the study area in the Review paper. I believe the authors could have already included the results of this current manuscript in that review paper to show an empirical example of vulnerability factors identified by other studies and avoid salami-slicing their well-presented research. I noticed the manuscript lacks a section detailing the design and classification process for the survey questions. This study could stand alone as a robust paper if it included a more sophisticated analysis in the questionnaire design phase. For example, the questions could be categorized independently of any prior knowledge of vulnerability factors. Methods such as PCA could then be applied to validate how respondents' answers align with the categories you've identified.

    *Response: Thank you for your comment regarding the contribution of our manuscript. We do not agree that our manuscript does not add much to our understanding of the vulnerability factors identified in our Review paper. Drought vulnerability is both complex and contextual, and there is a need to further our understanding of how traditional index-based approaches can incorporate multi-sectoral and multi-stakeholder perspectives. The approach taken in our manuscript involves stakeholders in several water dependent sectors, working in industry, authorities, NGOs, research, and trade associations. This provides a unique opportunity to investigate vulnerability from new perspectives stepping away from silo-approaches to drought risk assessments and management. We have, however, developed our introduction to better introduce the concept of indicator-based vulnerability assessments (Line 53-57, 74-80) and the role of expert knowledge (Line 80-93). As a result, the information on drought impacts in Sweden, previously included in the introduction, was moved to section 2.1 study area (Lines 157-173). We have also revised our Discussion to put our results in a broader context (e.g. Line 563-574, 590-598, 635-643).*
    *As for the section detailing the design and classification for the survey questions, 2.3.1 describes the design of the Survey. The respondents were presented with the vulnerability factors as a list, asking them to rate each factor on a scale from 0-4. However, to clarify the design of survey, we have now also included the survey questions as supplementary material (S8). Regarding the application of methods such as PCA, our study deliberately employs a multi-stage, expert-driven approach for factor identification and refinement. Given this structured methodology, we do not consider the inclusion of PCA necessary or appropriate within the scope of this manuscript.*

12. Section 3.1 can be summarized into one or two figures (pie chart, donut chart, bar chart, …). I find it redundant to report all the percentages of the answers in a text, mentioning only the important numbers is sufficient.

    *Response: We appreciate your suggestion. However, we find that the information presented in Section 3.1 is important for the overall understanding and interpretations of the results.*

13. Table 3: I believe it would make more sense if you reported the percentage instead of the number of respondents here. For example, for the environmental sector, 23% (8/35), 40% (14/35), and 37% (13/35) of the respondents have limited, moderate, and high drought experience respectively. I suggest this because the number of participants varies among sectors and it's hard to compare (similar to point size in Fig 2). Maybe, considering this suggestion not only in this table but throughout the manuscript could deliver more

understandable information, and could shorten your text by avoiding the reports of fractions of the total numbers (e.g., rating x out of y).

*Response: Thank you for your suggestions. The table has been revised to report the percentages instead of number of respondents (Line 330-334) and updated the text accordingly (Lines 321 & 324).*

14. Table 4: The order of presentation for the main 3 categories is also important to be consistent throughout the manuscript and in any supplementary materials, e.g., Sectoral, Societal, Governance.

*Response: Thank you for your comment. We have carefully gone through our original manuscript, and could not find any instances were the ordering of these categories was inconsistent.*

15. Section 3.2 is also a detailed report of all the numbers in Table 4. The table alone has all the information. I believe you could focus and elaborate only on important and meaningful numbers or percentages.

*Response: Thank you for your comment. We have revised section 3.2 as per your suggestion to elaborate on the significant findings of the section (Lines 354-361).*

16. Line 364: I suggest using the present verb form when reporting the rates: "tend"

*Response: We appreciate your suggestion. However, we believe that the section should be written in past tense as it involves the results obtained. We have used past-tense throughout the Results section of the manuscript, and believe that it is best to use the same tense for consistency.*

17. Line 366: "medium to high", or "medium and high"?

*Response: The sentence has been revised in the manuscript (Line 377).*

18. Section 3.3 is the most important part of your manuscript and is very well presented. I, particularly, like Fig 2 as it is very informative and comprehensive and summarizes your entire study at one look. However, I believe you can talk a lot more about these results. For instance, when it comes to the environmental sector which has a lot of participants, the impact score of susceptibility and coping capacity varies more among the factors compared to adaptive capacity.

*Response: Thank you for highlighting this finding as relevant. We have added this information to the text discussion (Line 382-385).*

19. Like Fig 2, Table 5 is also very informative and important to convey the message of this study.

*Response: Thank you for your kind words. We highly appreciate the feedback.*

20. Line 439, missing pronoun: "As "it" was seen …"

*Response: The sentence has been revised in the manuscript revised (Line 452)*

21. Section 3.5: For place of employment, generally, there were no significant differences between factor ratings. When it comes to governance factors, I understand why this difference does not exist (maybe because of the generally unified policies in Sweden and the low population density of the country). But can you elaborate on the potential reasons why for sectoral and societal factors, occasionally, there is no significant difference in the place of employment?

*Response: We agree that the results are interesting, and have revised our Discussion section to elaborate on potential reasons for this (Lines 751-764). We also added further results concerning differences in sectoral ratings based on place of employment in section 3.5 (Lines 480-486)*

22. Tables S6 and S7 are important parts of the study, and I believe that they can be summarized in two figures and shown in the main text.

*Response: We agree that the information in S6 and S7 are important. We have summarized the information in a new figure (Figure 4) and incorporated it in section 3.5 (Lines 513-519).*

23. Line 531-532: User validation => User-information

    *Response: See previous responses regarding the term "user-validation"*

24. Line 532-533: I suggest citing different articles here that assessed drought vulnerability instead of your own paper, which is a review, not an assessment.

    *Response: The cited article provides support to the claim that several studies have assessed drought vulnerability in the region. However, we acknowledge your comment and have revised the Discussion section. The original first paragraph of the discussion section has been rewritten in the revised manuscript to better describe the manuscripts main goal and key findings (Lines 551-558).*

25. Discussion: There are important messages pointed out by the authors in the Discussion, e.g., in Lines 584-585, they put the study results in a broader context and highlight the importance of climate region in the vulnerability factors assessment. However, most parts of the Discussion are a summary of the results that were already mentioned in the Results section or a self-citation to the authors' review paper. For example, in lines 589-596, the authors mention that there are significant differences in factor ratings between forestry and other sectors, "which may be an indication that such factors are of differing importance for the sector." Firstly, this is already mentioned in the Results section, and secondly, yes, a significant difference in factor ratings between sectors means different importance of that factor between the sectors which is not something new to us!

    *Response: We acknowledge that our Discussion section originally was too results-oriented. We have revised the Discussion to put our results in a broader context. The Discussion now involves discussion on the complexity (Lines 560-566) and contextuality of vulnerability (Lines 576 – 587, 589-606, 618-628, 635-643), both in relation to studies performed in other regions (Lines 563-598) and Nordic contexts (Lines 598-603). As well as the results implications for risk assessments (665-675, 685-691, 693-702), and policy developers (Lines 685-687, 732-741). We have also included a section on the application and limitations of participatory approaches (Lines 743-764).*

**2. Reviewer 2**

This article presents a survey with 102 respondents in different societal sectors in Sweden who are asked to rank drought vulnerability factors identified in a conceptual framework of the authors' design. The article is clearly structured and well written. Not being expert in quantitative social science methods I will not discuss the statistical analysis of the responses, but assuming that the calculations are correct the paper still has some major weaknesses in need of address before publication.

*Response: Thank you for assessing our manuscript and providing your detailed feedback.*

Firstly, further clarification of the logic underpinning the claim to have achieved user-validation of the drought vulnerability factors is needed. The authors present the result of a survey, this is an unusual approach to user-validation, a term currently commonly used in the context of software usability testing. In the software contexts it is clear what is meant by "user", but what do the authors of this paper mean? Also, is the term "validation" associated with the rather straight forward approach of software usability or with the philosophical discussion of validity as a criterium for claiming scientific truth? In addition to clarifying the semantics of the notion

*Response: Thank you for your comment. We realize that the term "user-validation" was unclear. We have now revised our manuscript and use the term "stakeholder perceptions" to describe our approach (Line 92,552, 618, 633, 645, 753).*

In-depth discussion about how the survey questions capture the perceptions of adequacy and appropriateness of the drought vulnerability factors among the selected users is also needed. It would be helpful to know what the questions in the survey were. The authors should also explain how inference was made from the survey responses to the validity of the drought vulnerability factors more extensively.

*Response: We have now included a section in the introduction describing the use of expert knowledge for weight assignment for index-based approaches (Lines 80-93). We have also revised our discussion to include discussion on participatory approaches (Lines 743-764). We now also include the survey questions in the supplementary materials (S8).*

Second, the authors must explain the contribution to knowledge intended by this article. The article is very descriptive, after stating that the purpose is to validate a particular conceptual framework for drought vulnerability the survey answers are accounted for in detail. But there is no explanation of why this is important to others than the authors.

*Response: Thank you for you comment. We have now revised our introduction to better describe the concept of indicator-based vulnerability assessments (Line 53-57, 74-80), the role of expert knowledge (Line 80-93), and the importance of our manuscript (Lines 92-103). We have also added a primary aim to the manuscript (Lines 101-103). Furthermore, we have revised our discussion to provide a more in-depth discussion on the manuscript's contribution to knowledge (e.g., Lines 570-574, 585-587, 590-598, 635-641, 646-652, 681-691).*

The lack of clarity about the contribution to knowledge becomes obvious in the discussion section where the authors summarise what has been presented in detail in the results and repeat what has already been said. There is no discussion about how the findings may impact on the existing body of knowledge. The authors should explain what this study means for the knowledge about drought vulnerability. What does the conceptual framework applied do that has not been done in previous research? Have they learnt anything that would challenge previous research findings? Does the study indicate that existing theory should be modified? What is the reader supposed to take away from this article? This must be stated much more clearly.

*Response: Thank you for your comments and suggestions. We have revised our Discussion section to better capture the manuscript's contribution to knowledge (e.g., e.g., Lines 570-574, 585-587, 590-598, 635-641, 646-652, 681-691) and have tried to address the questions you highlighted (e.g. Lines 671-675, 732-741).*

**3. Reviewer 3**

The manuscript investigates drought vulnerability factors identified in previous literature and claims to conduct a "user-validation" approach through a survey targeting stakeholders from several water-dependent sectors. The study explores an interesting and relevant subject, providing insights into the most critical drought vulnerability factors according to stakeholder perceptions. However, there are areas where the manuscript could benefit from significant improvement to enhance clarity, accessibility, and overall scientific impact. Below, I provide general and specific comments to help improve the manuscript.

*Response: Thank you for thoroughly reviewing our manuscript. We appreciate the time and effort. Please find the detailed responses to your comments below.*

**General Comments**

**Novelty and Scientific Contribution**

- Objective Clarity: The manuscript lacks a clear statement of its primary objective. The introduction suggests the paper provides a comprehensive analysis of drought vulnerability factors for water-dependent sectors, but this was already achieved in the previous paper (Stenfors et al., 2024). Based on the rest of the manuscript, the study appears to focus on user perceptions of these vulnerability factors rather than formal validation.

  - Recommendation: Revise the stated purpose of the work to reflect its actual contribution, emphasizing the user perspective as a preliminary step toward more elaborate validation efforts.

  *Response: Thank you for your comment regarding the lack of a clear primary objective and unclear distinction from our previous review paper. Please also refer to our comments to reviewer 1 and 2. As per your suggestion, we have now included a primary aim for our study (Line 101-103). We have also restructured our introduction, to put a larger emphasis on the importance of participatory approaches and expert knowledge for indicator-based vulnerability assessments (Lines 53-57, 74-80, 80-93). As a result, the information on drought impacts in Sweden, previously included in the introduction, was also moved to section 2.1 study area (Lines 157-173).*

- Terminology: The term "user-validation" is misleading. It may be more accurate to describe the work as "user perception analysis" of drought vulnerability factors. This distinction does not diminish the importance of the findings but ensures accuracy in describing the study's contribution.

  *Response: We acknowledge that the term "user-validation" was unclear, and have now rephrased it. In the manuscript, we now refer to our approach as a participatory approach using stakeholder perceptions approach (Line 92,552, 618, 633, 645, 753).*

- Overlap with Previous Work: The first stated objective, as identified in the introduction, repeats the findings of the previous paper. Consider removing or reframing it to avoid redundancy.

  *Response: Thank you for showing us that our first stated research objective could be misinterpreted as an overlap with our previous work. The first stated objective is not an overlap of our previous work. In our review-paper, we identified vulnerability factors that were described as relating to vulnerability, or used in vulnerability assessments in the region. However, the practical relevance of the identified factors for practitioners in the region was not further evaluated. We have now rephrased it, to distinguish it from our previous review-article (Lines 104-106).*

**Specific Comments Introduction**

1. Currency Conversion

- When using currency, such as SEK, convert it to USD for broader accessibility. Include a footnote with the conversion rate used, e.g., "Currency as of December 2024: 1 USD = 11 SEK" (adjust the exchange rate accordingly).

*Response: We have now changed the currency to USD and included a footnote with the exchange rate (Line 162 & 171).*

2. Conceptual Model

- Clarify why the conceptual model for drought vulnerability is critical. Include hints about its practical applications by addressing questions such as:

  - Where and how can this model be applied?

  - For whom is this model designed?

  A clear justification will help guide the readers through the manuscript's importance.

  *Response: We agree that the practical role of the conceptual model was insufficiently discussed in the first version of our manuscript. We have now revised our manuscript to provide more guidance as to the relevance and practical implications of our results and the conceptual model described (e.g. Line 668-675, 693-702, 732-741)*

**Methods**

3. Statistical Tests

- Provide references or examples of studies that used the Kruskal-Wallis and Wilcoxon Rank-Sum tests to justify their application. This is particularly helpful for readers unfamiliar with these methods.

  *Response: We have now added examples of three studies that used the applied statistical methods and added a short explanation of its relevance for non-parametric ordinal data. (Line 281-284),*

4. Regional Terminology Consistency

- Use consistent terminology when referring to the study region. For example, decide whether to refer to it as "Nordic countries" or "forested cold climates" and apply the same phrasing throughout.

  *Response: We have gone through the manuscript and revised this.*

**Discussion**

5. Depth of Discussion

- The discussion section currently reads as a summary of results. Include deeper analysis of the findings and their implications, specifically:

  - Why is the conceptual model important?

  - How can the user perspective on drought vulnerability factors contribute to water management or other real-world applications?

  - What distinguishes this study from previous research beyond the study region?

  *Response: Thank you for you comments and suggestions regarding our Discussion section. We have now revised and rewritten the discussion to provide a more in-depth analysis of our results, their implications (e.g. Lines 635-641, 668-671, 687-691,699-704), and contribution to knowledge (e.g. Lines 635-641, 646-652, 662-668).*

6. Comparisons to Previous Studies

- Instead of merely comparing the number of factors identified in other studies (e.g., Moshir Panahi et al., 2023; Ahmadalipour & Morakhani, 2018), highlight similarities or differences in specific vulnerability factors. This will add depth to the discussion.

  *Response: Thank you for your comment. We have revised the Discussion section to add more in-depth discussion on how our results compare to previous works and findings and their implications (e.g. Lines 563-568, 576-587, 592-606, 635-641, 678-681).*

7. Definitions of Key Terms

- Define terms such as adaptive capacity, coping capacity, and susceptibility, especially if they were included in the survey. Clarifying these terms will improve understanding and enhance the manuscript's accessibility.

  *Response: Thank you for your comment. However, we refer you to line 164-166 in the original manuscript for the definitions for the terms adaptive capacity, coping capacity, and susceptibility.*